# DOTA: DISTRIBUTIONAL TEST-TIME ADAPTATION OF VISION-LANGUAGE MODELS

## ABSTRACT

Vision-language foundation models (e.g., CLIP) have shown remarkable performance across a wide range of tasks. However, deploying these models may be unreliable when significant distribution gap exists between the training and test data. The training-free test-time dynamic adapter (TDA) is a promising approach to address this issue by storing representative test samples to guide the classification of subsequent ones. However, TDA only naively maintains a limited number of reference samples in the cache, leading to severe test-time catastrophic forgetting when the cache is updated by dropping samples. In this paper, we propose a simple yet effective method for DistributiOnal Test-time Adaptation (`Dota`). Instead of naively memorizing representative test samples, `Dota` continually estimates the distributions of test samples, allowing the model to continually adapt to the deployment environment. The test-time posterior probabilities are then computed using the estimated distributions based on Bayes' theorem for adaptation purposes. To further enhance the adaptability to uncertain samples, we introduce a new human-in-the-loop paradigm which identifies uncertain samples, collects human feedback, and incorporates it into the `Dota` framework. Extensive experiments validate that `Dota` enables model to continually learn during test-time, resulting in a significant improvement compared to state-of-the-art methods.

## 1 INTRODUCTION

Recent advances in vision-language foundation models have shown remarkable vision understanding capabilities across a broad range of tasks by training on web-scale image-text pairs (Radford et al., 2021; Lavoie et al., 2024; Zhai et al., 2023). Taking CLIP as an example, it can conduct zero-shot classification without the need for additional training data using predefined prompts (Radford et al., 2021). However, CLIP may still face challenges when handling various specific applications during test time, especially when there is a significant distribution gap between the training and test data (Shu et al., 2022; Karmanov et al., 2024; Feng et al., 2023).

Test-time adaptation methods are typically employed to address the distribution gap between the training and test datasets by fine-tuning the original model during test time (Boudiaf et al., 2022; Chen et al., 2022; Wang et al., 2021). Test-time adaptation aligns well with real-world applications where models need to adapt to new environments quickly. There are two primary lines to achieve test-time adaptation on the vision-language foundation models. Early works advocate learning prompts during test time with the test data (Shu et al., 2022; Feng et al., 2023). However, these methods require significant computational resources to optimize the learnable prompts via backpropagation and gradient descent. This significant resource overhead makes them unsuitable in applications when fast inference speed is widely required. Therefore, a more efficient method, Training-Free Dynamic Adapter (TDA), has been proposed (Karmanov et al., 2024) recently. To avoid the training process with backpropagation, TDA maintains a lightweight cache during testing to store representative test samples and guide the classification of subsequent test samples.

Although TDA has achieved significant efficiency compared to previous methods, it still faces challenges due to the limited cache capacity. Specifically, TDA naively preserves a limited number of typical samples in the cache during test time and dynamically updates the cache with higher classification-confidence samples. This strategy leads to test-time forgetting, because when new

confident samples are added, the previous cached samples must be discarded. As a result, relying solely on a few high-confidence samples stored in the cache may lead to a suboptimal classifier.

To address the above issue, we introduce a novel method called DistributiOnal Test-Time Adaptation (`Dota`). `Dota` continually estimates the distribution of test samples to adapt the test environment. Specifically, under the mild assumption that the embedding distribution of each class follows a Gaussian distribution (Hastie & Tibshirani, 1996), we propose an efficient method to continually estimate the distribution of different classes. Once the distributions of different classes are estimated, we can easily calculate the posterior probabilities of subsequent test samples based on Bayes' theorem and obtain a test-time classifier for test-time adaptation. Similar to TDA, this process does not require gradient backpropagation, avoiding the complex computational overhead during testing, leading to more than 20 times faster inference speed. Moreover, unlike TDA memorizing representative test samples, `Dota` can continually adapt to the test environment by estimating the distribution of different classes. Last but not least, to further improve the performance of the model in dealing with uncertain or risky samples during test-time adaptation, we introduce a new human-in-the-loop paradigm. This approach enables the model to detect uncertain samples and then adapt during test time with the aid of human feedback. This paradigm is crucial in scenarios where the model needs to adapt quickly to handle uncertainty during test-time. The contributions of this paper are:

- We propose a novel distributional test-time continual learning framework which improve the performance of existing visual-language foundation models in downstream tasks.
- Within this framework, we propose a simple yet effective method to enhance the foundation model by efficiently estimating the distribution of different categories during test time.
- We first define the test-time adaptation problem with human feedback, which allows the model to detect high-uncertainty samples and perform test-time adaptation under human feedback.
- Extensive experiments on diverse datasets validate the effectiveness of the proposed method, demonstrating a significant improvement. The code will be released for reproducing the results.

## 2 RELATED WORK

**Test-time adaptation (TTA)** focuses on addressing the distribution shift between training and test data by learning from the test data. Early efforts to improve TTA performance primarily involve adjusting batch normalization layers and designing unsupervised objective functions (Nado et al., 2020; Wang et al., 2020; Khurana et al., 2021; Lim et al., 2023). For example, TENT (Wang et al., 2020) optimizes the affine parameters in batch normalization layers by minimizing the entropy of the prediction probability. MEMO (Zhang et al., 2022a) applies variant augmentation methods to a single test sample and optimizes model parameters by minimizing the entropy of the prediction probability. T3A (Iwasawa & Matsuo, 2021) achieves test-time adaptation by adjusting the trained linear classifier using prototypes. To enhance the performance of vision-language models during testing, TPT (Shu et al., 2022) introduces adaptive text prompts and optimizes the prompts through entropy minimization. Building on this, DiffTPT (Feng et al., 2023) leverages pre-trained stable diffusion models to generate diverse augmented data for use in test-time prompt tuning. However, TPT and DiffTPT rely heavily on gradient backpropagation to optimize the prompts, making them computationally expensive and resource-intensive during testing. TDA (Karmanov et al., 2024) proposes a lightweight test-time adaption method by storing representative test samples. To enable practical test-time adaptation in dynamic, time-correlated test data streams, such as autonomous driving, RoTTA introduces novel robust batch normalization, a memory bank for balanced sampling, and a time-aware reweighting strategy(Yuan et al., 2023). A recent advancement in test-time adaptation with distribution shift, which introduces the concept of universal TTA to address domain non-stationarity and temporal correlation, ensuring robust model performance across diverse scenarios (Marsden et al., 2024). Compared to TDA and T3A, which naively stores typical test samples, we achieve continuous adaptation by estimating the distribution of test samples, leading to a more efficient and adaptive solution.

**Distribution estimation for recognition.** Distribution estimation leverages statistical properties of data to dynamically update models, enabling effective recognition in scenarios with new classes or shifting distributions(Hastie & Tibshirani, 1996). For instance, Bendale & Boult (2015) introduces a recognition system capable of continuously learning new object categories within an open-world

framework by extending nearest class mean algorithms into a nearest non-outlier (NNO) algorithm. Snell et al. (2017) propose prototypical networks, which leverage distribution estimation by representing each class with a prototype computed as the mean of embedded support points, enabling classification through metric space distances and achieving excellent results in few-shot and zero-shot learning scenarios. De Lange & Tuytelaars (2021) introduce a system for continual prototype evolution, enabling online learning and prediction from non-stationary data streams through efficient memory schemes and a novel objective function. In this paper, `Dota` revisits principles in the literature on continual learning via nearest class mean classifiers (Bendale & Boult, 2015; Mensink et al., 2013) for improving the performance of vision-language models at test time with human feedback.

**Uncertainty estimation** aims to estimate the reliability of decision. Traditional methods for uncertainty estimation often require additional training processes. For example, ensemble learning (Lakshminarayanan et al., 2017; Liu et al., 2019) and Bayesian neural networks (MacKay, 1992; Gal & Ghahramani, 2016) estimate uncertainty by obtaining the distribution of prediction. However, these methods typically introduce additional computational costs during inference. To address this, regularization-based methods have been proposed to constrain the confidence of the model during training, preventing overfitting and thereby improving uncertainty estimation (Malinin & Gales, 2018; Sensoy et al., 2018; Han et al., 2022; 2024). However, these methods focus on modifying the training process, such as changing the model architecture or loss function, to estimate uncertainty. They are not applicable to foundation models that have already been fully trained. Therefore, in this paper, we focus on estimating uncertainty during the inference stage using test samples.

**Vision-language models** have demonstrated strong vision understanding capabilities benefiting from training on large-scale datasets (Radford et al., 2021; Zhai et al., 2023; Lavoie et al., 2024). Among them, CLIP (Radford et al., 2021) is the most representative method by maximizing the similarity between image and their corresponding text embeddings. To further enhance performance of CLIP on downstream tasks, prompt learning-based methods have been proposed by optimizing the prompts of the text encoder (Zhou et al., 2022a;b; Bai et al., 2024; Khattak et al., 2023). Moreover, to reduce the computational cost associated with gradient calculations in prompt learning, efficient CLIP adaptation methods have been introduced (Gao et al., 2024; Zhang et al., 2022b; Wang et al., 2024; Li et al., 2024; Yu et al., 2023). These methods enable downstream task adaptation using only a small number of training samples in the embedding space. Orthogonal to above methods, this paper focuses on continuously adapting to environments during testing by leveraging test samples.

## 3 METHOD

**Zero-shot classification.** During the pre-training stage, CLIP[1] trains its image and text encoders using large-scale image-text pairs. This is achieved by maximizing the cosine similarity between the image and text embeddings through contrastive loss. Unlike traditional classifiers trained on closed-set labels, CLIP leverages open-set semantic information in the image-text pairs to learn a broader range of visual concepts. Consequently, during the test stage, CLIP can perform zero-shot classification without additional training. Specifically, given a test sample $\boldsymbol{x}$ for $K$-class classification, where $\boldsymbol{x}$ represents the image embedding obtained from the image encoder, the corresponding zero-shot prediction probability $P_k^{\mathrm{zs}}$ for class $k$ is calculated as:

$$P_k^{\mathrm{zs}}(y = k|\boldsymbol{x}) = \frac{\exp(\cos(\boldsymbol{x}, \boldsymbol{w}_k)/\tau)}{\sum_{k=1}^{K} \exp(\cos(\boldsymbol{x}, \boldsymbol{w}_k)/\tau)}, \tag{1}$$

where $\mathrm{zs}$ refers to **z**ero-**s**hot. $\boldsymbol{w}_k$ is the classification weight for class $k$, obtained by encoding the corresponding prompt, e.g., "a photo of {class}", with the class token replaced by the specific category name. $\tau$ is the learned temperature parameter in CLIP, and $\cos(\cdot, \cdot)$ denotes the cosine similarity. The above classification process can be understood as comparing the obtained image embedding with the text prompt and selecting the most similar category as the final decision.

### 3.1 DISTRIBUTIONAL TEST-TIME ADAPTATION

**Motivation.** When CLIP is deployed in various environments, the performance tends to degrade due to the changes of data distribution, especially when the test data has a significant distribution gap from the CLIP training data. Test-time adaptation can effectively adapt the foundational model

---

[1]While this paper primarily focuses on CLIP, our approach is also applicable to other similar models.

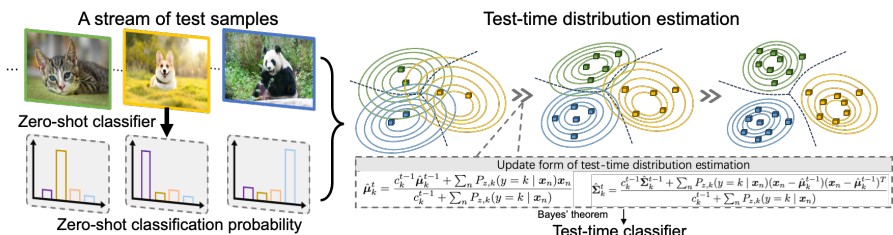

Figure 1: Pipeline of the proposed method. During test time, a stream of test samples is evaluated with original zero-shot classifier, and we estimate the distributions for the test samples during testing, enabling the model to continually learn from the test samples and the zero-shot classification probabilities. *As the number of test samples increases, the estimated test sample data distribution will become more accurate.* Finally the test-time classifier can then be obtained using the estimated distributions according to Bayes' theorem for test-time adaptation.

to new environments quickly during the test stage. Current state-of-the-art method TDA maintains a cache during test-time to preserve representative samples of different classes, which then guide the classification of the following test samples. However, TDA may lead to a severe test-time forgetting problem when the cache is updated due to only maintaining the embeddings of very limited test samples without learning the underlying relationships between the sample and label. To this end, we propose distributional test-time adaptation (`Dota`), which aims to continuously learn from test-time data by estimating the test sample distribution. Specifically, as shown in Fig. 1, we propose to online estimate the data distribution of samples in the current test environment during testing. Once obtaining the distribution, we can leverage Bayes' theorem to naturally infer the test-time posterior distribution of different classes for new test samples to adapt the test-time environment.

**Classification with classical Gaussian discriminant analysis.** Formally, inspired by classical Gaussian discriminant analysis (Hastie & Tibshirani, 1996), we assume that the embedding distribution of each class $k$ follows a Gaussian distribution, i.e., $P(\boldsymbol{x}|y{=}k) = \mathcal{N}(\boldsymbol{\mu}_k, \boldsymbol{\Sigma}_k)$, where $\boldsymbol{\mu}_k$ and $\boldsymbol{\Sigma}_k$ are the mean vector and covariance matrix of class $k$, respectively. Using Bayes' theorem, the posterior probability $P(y=k|\boldsymbol{x})$ of class $k$ can be given by $P(y=k|\boldsymbol{x}) = \frac{P(\boldsymbol{x}|y{=}k)P(y{=}k)}{P(\boldsymbol{x})}$, where $P(\boldsymbol{x}) = \sum_{k=1}^{K} P(\boldsymbol{x}|y=k)P(y=k)$ and $P(y=k)$ is the prior probability. In practice, we set the prior probability to $1/K$ for simplicity. Then $P(y=k|\boldsymbol{x})$ can be obtained with

$$P(y= k \mid \boldsymbol{x}) = \frac{\exp(f_k(\boldsymbol{x}))}{\sum_{k=1}^{K} \exp(f_k(\boldsymbol{x}))}, \tag{2}$$

where $f_k(\boldsymbol{x}) = -\frac{1}{2}(\boldsymbol{x}-\boldsymbol{\mu}_k)^T \boldsymbol{\Sigma}_k^{-1}(\boldsymbol{x}-\boldsymbol{\mu}_k) - \frac{1}{2}\log|\boldsymbol{\Sigma}_k|$. The discriminant function $f_k(\boldsymbol{x})$ measures how well a sample $\boldsymbol{x}$ fits the distribution of class $k$. The detail can be found in the Appendix A.2.

**Parameter estimation with zero-shot predictive probability.** We can conduct classifier updating with the Gaussian discriminant analysis. Unfortunately, during testing, we cannot access to the ground-truth labels for the $N$ test samples, whose input embeddings are denoted as $\{\boldsymbol{x}_n\}_{n=1}^{N}$. Therefore, we try to use the zero-shot predictive probability to estimate the distribution (Hastie & Tibshirani, 1996). Specifically, we first estimate the zero-shot posterior probability $\{P_k^{zs}\}_{k=1}^{K}$. Then, we maximize the likelihood by estimating the means $\{\hat{\boldsymbol{\mu}}_k\}_{k=1}^{K}$ and covariances $\{\hat{\boldsymbol{\Sigma}}_k\}_{k=1}^{K}$. This process can be viewed as a single iteration of the EM algorithm (Moon, 1996), where obtaining the zero-shot classification probability corresponds to the expectation step, and estimating $\{\hat{\boldsymbol{\mu}}_k, \hat{\boldsymbol{\Sigma}}_k\}_{k=1}^{K}$ based on the zero-shot predicted probability corresponds to the maximization step, adhering to the principle of maximum likelihood estimation. Formally, $\{\hat{\boldsymbol{\mu}}_k, \hat{\boldsymbol{\Sigma}}_k\}_{k=1}^{K}$ can be estimated with:

$$\hat{\boldsymbol{\mu}}_k = \frac{\sum_{n=1}^{N} P_k^{zs}(y{=}k|\boldsymbol{x}_n)\boldsymbol{x}_n}{\sum_{n=1}^{N} P_k^{zs}(y{=}k|\boldsymbol{x}_n)}, \quad \hat{\boldsymbol{\Sigma}}_k = \frac{\sum_{n=1}^{N} P_k^{zs}(y{=}k|\boldsymbol{x}_n)(\boldsymbol{x}_n-\hat{\boldsymbol{\mu}}_k)(\boldsymbol{x}_n-\hat{\boldsymbol{\mu}}_k)^T}{\sum_{n=1}^{N} P_k^{zs}(y{=}k|\boldsymbol{x}_n)}. \tag{3}$$

The above estimation can also be intuitively understood as reweighting, where the zero-shot predicted probabilities are used as weights to adjust the contributions of different samples, thereby mitigating the impact of the potential inaccuracies in the zero-shot predicted probabilities.

**Test-time distribution estimation.** When estimating data distribution at test time, one another challenge is that we evaluate the test samples sequentially in a streaming manner instead of accessing all samples simultaneously. This necessitates a strategy to appropriately adjust the estimation method in Eq. 3 through effective initialization, and then allowing the parameters to be updated quickly as new test samples arrive. To achieve this goal, Dota maintains the distribution information of different classes (i.e., mean and covariance matrix) during testing, and updates its distribution information based on its representation information after obtaining new samples. **Initialization of** $\{\hat{\boldsymbol{\mu}}_k, \hat{\boldsymbol{\Sigma}}_k\}_{k=1}^K$. We can initialize the estimated mean of different classes in a way that aligns it with the original zero-shot classifier $\{\boldsymbol{w}_k\}_{k=1}^K$, i.e., $\hat{\boldsymbol{\mu}}_k^0 = \boldsymbol{w}_k$ and $\hat{\boldsymbol{\Sigma}}_k^0 = \sigma^2 \boldsymbol{I}$, where $\sigma^2$ is a hyperparameter that determines the initial variance and $\boldsymbol{I}$ is the identity matrix. **Update of** $\{\hat{\boldsymbol{\mu}}_k, \hat{\boldsymbol{\Sigma}}_k\}_{k=1}^K$. We employ the update form described in (Dasgupta & Hsu, 2007), which is capable of estimating Gaussian distribution parameters in an online setting. Theoretically, for any sequence, the average regret of the update form converges to zero in the limit. Specifically, given a batch of test samples at step $t$, the updated $\hat{\boldsymbol{\mu}}_k^t$, $\hat{\boldsymbol{\Sigma}}_k^t$ can be computed based on the $\hat{\boldsymbol{\mu}}_k^{t-1}$, $\hat{\boldsymbol{\Sigma}}_k^{t-1}$ as follows:

$$\hat{\boldsymbol{\mu}}_k^t = \frac{c_k^{t-1}\hat{\boldsymbol{\mu}}_k^{t-1} + \sum_n P_k^{\text{zs}}(y=k|\boldsymbol{x}_n)\boldsymbol{x}_n}{c_k^{t-1} + \sum_n P_k^{\text{zs}}(y=k|\boldsymbol{x}_n)} \text{ and } \hat{\boldsymbol{\Sigma}}_k^t = \frac{c_k^{t-1}\hat{\boldsymbol{\Sigma}}_k^{t-1} + \sum_n P_k^{\text{zs}}(y=k|\boldsymbol{x}_n)(\boldsymbol{x}_n - \hat{\boldsymbol{\mu}}_k^{t-1})(\boldsymbol{x}_n - \hat{\boldsymbol{\mu}}_k^{t-1})^T}{c_k^{t-1} + \sum_n P_k^{\text{zs}}(y=k|\boldsymbol{x}_n)},$$
(4)

where $c_k^{t-1}$ is the sum of the confidences of the cumulative number of observed samples of class $k$ at step $t-1$, and $c_k^0 = 1$, with $c_k^t$ updated as $c_k^t = c_k^{t-1} + \sum_n P_k^{\text{zs}}(y=k|\boldsymbol{x}_n)$. Then, we can use Eq. 2 to calculate the test-time adapted posterior probability. In practice, Eq. 4 is a generalized vector update version that works effectively with different test batch sizes. For consistency with comparison methods, we set the batch size to 1 in our experiments. To reduce computational complexity when inverting the covariance matrix $\hat{\boldsymbol{\Sigma}}_k$, similar to the approach in (Anderson et al., 1958; Friedman, 1989), we approximate the covariance by averaging across all classes, reducing the number of matrix inversions from $K$ to 1, thereby improving efficiency. Additionally, we apply shrinkage regularization to the precision matrix to enhance the stability of the inversion process as follows: $\hat{\boldsymbol{\Lambda}} = [(1-\epsilon)\hat{\boldsymbol{\Sigma}} + \epsilon\boldsymbol{I}]^{-1}$, where $\epsilon = 10^{-4}$ is the shrinkage parameter. The term $\epsilon\boldsymbol{I}$ ensures that the eigenvalues of the covariance matrix are well-conditioned, maintaining the desired properties such as positive definiteness and rank stability.

**Comparison with single image TTA(Khurana et al., 2021)**. In single image TTA, the model makes predictions based solely on the given test instance. However, Dota is a versatile TTA method that works seamlessly in both single-image and multi-image settings benefiting from its vectorized distribution estimation strategy and preserving class means and covariance matrix parameters.

### 3.2 TEST-TIME ADAPTION WITH HUMAN FEEDBACK

**Test-time adaption with human feedback.** Dota enhances model performance by estimating the data distribution of incoming test samples. However, relying solely on zero-shot predicted probability distributions for this estimation may lead to inaccuracies, particularly for originally uncertain samples. The predicted probabilities of these uncertain samples often fail to provide reliable information for accurate distribution estimation. To address this, we propose a new task that incorporates human feedback during test-time adaptation, establishing a simple yet effective human-in-the-loop paradigm. Specifically, after the model is deployed, we aim to obtain label information on uncertain samples with human in real-time and use it for test-time adaptation. This approach enables quick and effective performance improvements on uncertain samples during testing.

**Test-time uncertainty estimation.** To achieve the test-time adaption with human feedback, we first define the test-time uncertainty estimation task, which aims to determine whether the current test sample is uncertain based

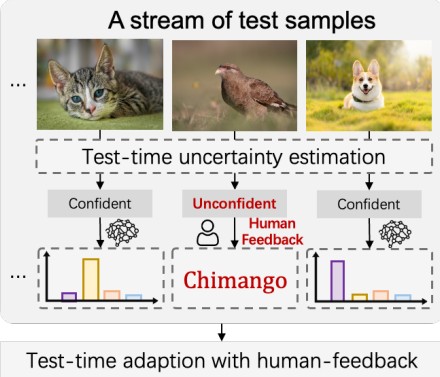

Figure 2: Test-time uncertainty estimation is employed to identify unconfident samples, prompting the input of human feedback. The feedback, combined with the prediction of model, is then utilized for test-time adaptation.

on the information from the previous test samples stream. Formally, given a test sample $\boldsymbol{x}_i$ and the previously tested samples $\{\boldsymbol{x}_n\}_{n=1}^{i-1}$, our objective is to evaluate whether the current sample $\boldsymbol{x}_i$ is uncertain, leveraging information from both the previous inference samples $\{\boldsymbol{x}_n\}_{n=1}^{i-1}$ and $\boldsymbol{x}_i$ itself. To achieve this goal, we propose a simple yet effective method based on the confidence scores of past samples[2]. Specifically, we store the confidence scores of all past test samples and use this information to determine whether the current test sample falls within the lowest percentile of confidence scores. Formally, given the confidence score $s_i$ of the current sample $\boldsymbol{x}_i$, where $s_i = \max(\{P_k^{\mathrm{zs}}(y=k|\boldsymbol{x}_i)\}_{k=1}^K)$, we classify $\boldsymbol{x}_i$ as uncertain if $s_i \leq s_\gamma$. where $s_\gamma = \mathrm{percentile}(\{s_n\}_{n=1}^i, \gamma)$ represents the value at the $\gamma$-th percentile of the confidence scores $\{s_n\}_{n=1}^i$, with $\gamma$ indicating the proportion of scores when sorted in ascending order. In other words, $s_\gamma$ corresponds to the score below which $\gamma$ proportion of the sorted confidence scores fall. Moreover, $\gamma$ can be viewed as a hyperparameter that controls the proportion of samples classified as uncertain, which can be used to control the degree of human involvement during the testing process. Compared with the traditional method of judging whether the decision is uncertain only based on the current sample, we can obtain relative uncertainty estimation to improve the adaptability of model to the test data distribution and more robust threshold setting. Then as shown in Fig. 2 when sample is uncertain, we can collect human feedback, manually determine its true label, and use the method in Sec. 3.1 to continuously update the model. **Why confidence from zero-shot classifier.** The estimated confidence is derived from a zero-shot classifier because the pre-trained CLIP model demonstrates strong calibration in the zero-shot setting (Minderer et al., 2021). We can also use other calibration methods (Tu et al., 2024; Wang et al.) to further improve the reliability of confidence.

**Difference between test-time adaption with human feedback and active learning.** Active learning (Holub et al., 2008; Sener & Savarese, 2018; Ash et al., 2020; Bang et al., 2024) is a paradigm where a model selects the most informative samples for labeling to optimize learning efficiency. The core difference in sample selection strategies between active learning and test-time adaptation with human feedback lies in the data availability setting. Specifically, active learning assumes simultaneous access to a small labeled dataset and the entire pool of unlabeled data, enabling the scoring and selection of the most valuable samples for labeling from the complete dataset. In contrast, test-time adaptation with human feedback processes a continuous stream of data, where each sample is presented sequentially and cannot be revisited later. This necessitates immediate, on-the-fly decisions about collecting human feedback based solely on the current sample and insights from previously observed samples, without prior access to the entire dataset.

**Similarity between test-time adaptation with human-feedback and active learning.** Both active learning and test-time adaptation with human feedback involve scoring samples to assess their value, using criteria such as uncertainty (Nguyen et al., 2022), diversity (Sener & Savarese, 2018; Ash et al., 2020), or confidence (Li & Sethi, 2006). In this work, we adopt a simple confidence-based scoring method to decide whether to collect human feedback. We also explore other criteria, such as similarity-based scoring in our experiments.

## 3.3 ADAPTIVE FUSION OF ZERO-SHOT AND TEST-TIME CLASSIFIER

As the number of test samples increases, the reliability of the estimated test sample distribution improves (Dasgupta & Hsu, 2007). However, when the number of test samples is insufficient, the estimated distribution may be unreliable. To address this, we introduce a dynamic zero-shot classification and test-time result fusion approach, allowing the model to rely more on zero-shot classification during stages where the sample size for distribution estimation is insufficient. Formally, the final fusion probability is defined as follows:

$$P_k(y = k|x) = \frac{\exp(\cos(\boldsymbol{x}, \boldsymbol{w}_k)/\tau + \lambda f_k(\boldsymbol{x}))}{\sum_{k=1}^K [\exp(\cos(\boldsymbol{x}, \boldsymbol{w}_k)/\tau + \lambda f_k(\boldsymbol{x}))]}, \tag{5}$$

where $\lambda = \min(\rho c, \eta)$. Here, $c$ represents the number of test samples, and $\rho$ and $\eta$ are hyperparameters that control the weight of the test-time classifier logits. The value of $\lambda$ increases with the number of test samples when this number is insufficient, gradually approaching the maximum value $\eta$. This approach encourages the model to rely on the zero-shot classifier results when the test samples are insufficient to estimate the distribution, mitigating the potential negative impact of the test-time classifier. The whole pseudo code is shown in Alg. 1.

---

[2]We propose a simple yet effective solution and leave the task of improving performance to future work.

## 4 EXPERIMENTS

**Benchmarks.** Consistent with prior works (Shu et al., 2022; Feng et al., 2023; Karmanov et al., 2024), we conduct our main experiments on natural distribution shifts and cross-domain generalization scenarios. For the natural distribution shifts scenario, we utilize multiple datasets including ImageNet (Deng et al., 2009), ImageNet-A (Hendrycks et al., 2021b), ImageNet-R (Hendrycks et al., 2021a), ImageNetV2 (Recht et al., 2019), and ImageNet-S (Wang et al., 2019), which serve as measures of the robustness of our approach. In the cross-domain generalization scenario, we evaluate the performance of the model across 10 diverse image classification datasets, each representing a distinct domain with different classes: Aircraft (Maji et al., 2013), Caltech101 (Fei-Fei et al., 2004), Cars (Krause et al., 2013), DTD (Cimpoi et al., 2014), EuroSAT (Helber et al., 2019), Flower102 (Nilsback & Zisserman, 2008), Food101 (Bossard et al., 2014), Pets (Parkhi et al., 2012), SUN397 (Xiao et al., 2010), and UCF101 (Soomro et al., 2012). This benchmark provides a comprehensive evaluation of the adaptability of the model during test time across various class spaces.

**Evaluation metrics.** When there is no human feedback, We report the accuracy of different methods. When human feedback is available, we evaluate performance using two metrics: standard accuracy (ACC) and feedback-enhanced accuracy (ACC$^\star$). ACC evaluates accuracy using the predicted labels before incorporating human feedback, ensuring comparability with other methods by using the same number of test samples. In contrast, ACC$^\star$ uses the updated labels for samples with human feedback to evaluate the overall accuracy, highlighting the benefit of incorporating feedback.

**Choice of hyperparameters.** To ensure a fair comparison with other methods, we used the same experimental settings, adjusting model hyperparameters based on the validation set. However, we repeated the experiments and found that the proposed method is inherently robust to hyperparameter variations, achieving strong performance on the test set without hyperparameters tuning.

**Comparison method.** We compare the proposed method with TPT (Shu et al., 2022), DiffTPT (Feng et al., 2023), ZERO (Farina et al., 2024), TDA (Karmanov et al., 2024), BoostAdapter (Zhang et al., 2024b) and HisTPT(Zhang et al., 2024a). To be consistent with the previous works (Karmanov et al., 2024), we also include the baseline zero-shot performance of CLIP, using the ensemble of 80 hand-crafted prompts (Radford et al., 2021). We compare with ATPT(Sarkar et al., 2024) that also incorporates human feedback.

### 4.1 COMPARISON WITH STATE-OF-THE-ARTS METHODS

| Method | BP-free | Continual adaption | ImageNet | ImageNet-A | ImageNet-R | ImageNet-S | Average | ImageNetV2 |
|---|---|---|---|---|---|---|---|---|
| CLIP-ViT-B/16 | ✓ | ✗ | 68.34 | 49.89 | 77.65 | 48.24 | 61.03 | 61.88 |
| TPT | ✗ | ✗ | 68.98 | 54.77 | 77.06 | 47.94 | 62.19 | 63.45 |
| DiffTPT | ✗ | ✗ | 70.30 | 55.68 | 75.00 | 46.80 | 61.95 | 65.10 |
| TDA | ✓ | ✗ | 69.51 | 60.11 | 80.24 | 50.54 | 65.10 | 64.67 |
| Dota | ✓ | ✓ | **70.68** | **61.19** | **81.17** | **51.33** | **66.09** | 64.41 |
| Dota 5% feedback | ✓ | ✓ | 71.01 | 61.44 | 81.41 | 52.13 | 66.50 | 64.45 |
| Dota 5% feedback$^\star$ | ✓ | ✓ | 74.52 | 64.72 | 85.01 | 55.99 | 70.06 | 68.11 |
| Dota 15% feedback | ✓ | ✓ | 71.83 | 61.83 | 81.78 | 53.34 | 67.20 | 64.53 |
| Dota 15% feedback$^\star$ | ✓ | ✓ | 80.91 | 71.33 | 90.15 | 64.18 | 76.64 | 75.38 |
| CLIP-ResNet-50 | ✓ | ✗ | 59.81 | 23.24 | 60.72 | 35.48 | 44.81 | 52.91 |
| TPT | ✗ | ✗ | 60.74 | 26.67 | 59.11 | 35.09 | 45.40 | 52.91 |
| DiffTPT | ✗ | ✗ | 60.80 | **31.06** | 58.80 | 37.10 | 46.94 | 55.80 |
| TDA | ✓ | ✗ | 61.35 | 30.29 | 62.58 | **38.12** | 48.09 | 55.54 |
| Dota | ✓ | ✓ | **61.82** | 30.81 | **62.81** | 37.52 | **48.24** | 55.27 |
| Dota 5% feedback | ✓ | ✓ | 62.12 | 31.01 | 63.04 | 37.86 | 48.51 | 55.30 |
| Dota 5% feedback$^\star$ | ✓ | ✓ | 65.92 | 35.32 | 67.42 | 42.31 | 52.74 | 59.05 |
| Dota 15% feedback | ✓ | ✓ | 62.77 | 31.13 | 63.34 | 38.48 | 48.93 | 55.34 |
| Dota 15% feedback$^\star$ | ✓ | ✓ | 73.22 | 43.66 | 74.98 | 51.07 | 60.73 | 66.51 |

Table 1: Top-1 accuracy and accuracy with human feedback(with $^\star$)(%) under the natural distribution shifts scenario. For clarity, the best and second-best results that do not require human-feedback are shown in **bold** and underlined, respectively. Dota 5% and 15% feedback indicate test-time adaptation with human-feedback on uncertain samples, with approximately 5% and 15% of the samples being uncertain ($\gamma = 0.05$ or $0.15$). BP-free and continual adaption indicate whether the method does not require gradient backpropagation and has the ability of continuous adaptation. Last column shows the failure cases of Dota. Detailed reasons are in the experimental results section.

**Results under the natural distribution shifts scenario and the cross-domain generalization scenario.** We first compare Dota with state-of-the-art methods in the context of natural distribution shifts. Tab. 1 and Tab. 3 present the experimental results, revealing several key observations. (1) Leveraging distribution modeling of the representation of test data, Dota achieves superior performance without requiring gradient backpropagation. (2) Performance of Dota can be further improved by incorporating human feedback. For example, with the ViT-B/16 backbone, introducing human feedback for approximately 5% of uncertain inference samples during test-time adaptation leads to an additional average performance improvement of 0.41%. When the collected human feedback was used to replace the model's original predictions, model performance was further significantly improved. (3) The performance improvement achieved by DOTA on ResNet-50 is notably smaller compared to ViT-B/16. This discrepancy can be attributed to differences in representation dimensions. Specifically, ResNet's representation dimension is 1024, while ViT-B/16's is 512. In our method, this results in a significant increase in the number of parameters required to estimate the test data distribution. (4) Dota achieves better performance with human feedback. As shown in Tab. 3, compared to TDA and ATPT, Dota achieves an average performance of 70.96 with 5% feedback, while ATPT and TDA achieve 67.26 and 65.31, respectively. (5) While our approach demonstrates the advantage of continuously estimating the distribution of test data, allowing for adaptation to test data, it does not consistently outperform TDA on all the dataset. For example, as shown in Tab. 1, on ImagenetV2 datasets with only 10 samples per class, Dota does not significantly exceed TDA.

| Method | Kather | PanNuke | WSSS4LUAD | Average |
|---|---|---|---|---|
| PLIP (Baseline) | 45.60 | 70.31 | | 62.49 |
| TDA | 49.39 | 71.56 | 72.13 | 64.36 |
| DOTA | 55.22 | 72.25 | 72.32 | 66.60 |
| 5% Human Feedback | 56.52 | 72.35 | 72.62 | 67.16 |
| 5% Human Feedback* | 57.82 | 73.83 | 73.25 | 68.30 |
| 15% Human Feedback | 58.32 | 72.46 | 72.79 | 67.86 |
| 15% Human Feedback* | 61.60 | 76.91 | 74.41 | 70.97 |

Table 2: Comparisons of PLIP and proposed methods across medical datasets.

| Method | Aircraft | Caltech101 | Cars | DTD | EuroSAT | Flower102 | Food101 | Pets | SUN397 | UCF101 | Average |
|---|---|---|---|---|---|---|---|---|---|---|---|
| CLIP-ViT-B/16 | 23.22 | 93.55 | 66.11 | 45.04 | 50.42 | 66.99 | 82.86 | 86.92 | 65.63 | 65.16 | 64.59 |
| TPT | 24.78 | 94.16 | 66.87 | 47.75 | 42.44 | 68.98 | 84.67 | 87.79 | 65.50 | 68.04 | 65.10 |
| DiffTPT | 25.60 | 92.49 | 67.01 | 47.00 | 43.13 | 70.10 | 87.23 | 88.22 | 65.74 | 62.67 | 65.47 |
| TDA | 23.91 | 94.24 | 67.28 | 47.40 | 58.00 | 71.42 | 86.14 | 88.63 | 67.62 | 70.66 | 67.53 |
| ZERO | 25.21 | 93.66 | 68.04 | 46.12 | 34.33 | 67.68 | 86.53 | 87.75 | 65.03 | 67.77 | 64.21 |
| BoostAdapter | 27.45 | 94.77 | 69.30 | 45.69 | 61.22 | 71.66 | 87.17 | 89.51 | 68.09 | 71.93 | 68.68 |
| HisTPT | 26.90 | 94.50 | 69.20 | 48.90 | 49.70 | 71.20 | 89.30 | 89.10 | 67.20 | 70.10 | 67.60 |
| Dota | 25.59 | 94.32 | 69.48 | 47.87 | 57.65 | 74.67 | 87.02 | 91.69 | 69.70 | 72.06 | 69.01 |
| ATPT 5% feedback | 24.85 | 94.27 | 67.86 | 48.23 | 49.88 | 72.36 | 86.77 | 90.65 | 67.51 | 70.23 | 67.26 |
| TDA 5% feedback | 23.13 | 91.36 | 64.73 | 41.78 | 55.54 | 69.47 | 85.87 | 89.48 | 64.54 | 67.17 | 65.31 |
| Dota 5% feedback | 26.73 | 94.56 | 70.95 | 49.82 | 65.00 | 76.86 | 87.17 | 92.78 | 70.49 | 75.26 | 70.96 |
| TDA 15% feedback | 23.73 | 91.93 | 66.02 | 44.27 | 64.06 | 70.52 | 85.97 | 90.52 | 65.8 | 71.56 | 67.44 |
| Dota 15% feedback | 28.65 | 95.01 | 73.01 | 53.78 | 76.60 | 79.70 | 87.41 | 93.54 | 71.82 | 79.33 | 73.89 |
| CLIP-ResNet-50 | 16.11 | 87.26 | 55.89 | 40.37 | 25.79 | 62.77 | 74.82 | 82.97 | 60.85 | 59.48 | 56.63 |
| TPT | 17.58 | 87.02 | 58.46 | 40.84 | 28.33 | 62.69 | 74.88 | 84.49 | 61.46 | 60.82 | 57.66 |
| DiffTPT | 17.60 | 86.89 | 60.71 | 40.72 | 41.04 | 63.53 | 79.21 | 83.40 | 62.72 | 62.67 | 59.85 |
| TDA | 17.61 | 89.70 | 57.78 | 43.74 | 42.11 | 68.74 | 77.75 | 86.18 | 62.53 | 64.18 | 61.03 |
| HisTPT | 18.10 | 87.20 | 61.30 | 41.30 | 42.50 | 67.60 | 81.30 | 84.90 | 63.50 | 64.10 | 61.20 |
| Dota | 18.06 | 88.84 | 58.72 | 45.80 | 47.15 | 68.53 | 78.61 | 87.33 | 63.89 | 65.08 | 62.20 |
| TDA 5% feedback | 15.75 | 84.91 | 54.47 | 37.77 | 48.86 | 64.43 | 76.66 | 82.53 | 57.7 | 60.48 | 58.36 |
| Dota 5% feedback | 18.81 | 89.25 | 59.22 | 47.10 | 59.36 | 69.63 | 78.75 | 84.65 | 64.65 | 68.04 | 64.31 |
| TDA 15% feedback | 16.05 | 85.52 | 56.17 | 41.02 | 55.1 | 65.81 | 76.87 | 84.41 | 59.24 | 63.05 | 60.32 |
| Dota 15% feedback | 19.62 | 89.98 | 60.34 | 51.83 | 68.19 | 72.59 | 79.06 | 88.96 | 65.96 | 72.46 | 66.90 |

Table 3: Top-1 accuracy (%) under the cross-domain generalization scenario.

**Results under other dataset and CLIP-like foundation model.** We conducted experiments using PLIP (Huang et al., 2023) as the backbone and baseline for comparison, evaluating our method on three medical image datasets Kather, PanNuke, and WSSS4LUAD. The results, summarized in Tab. 2, show that DOTA consistently outperformed other methods, with further accuracy improvements observed when incorporating human feedback.

| Method | Testing Time | Accuracy | Gain |
|---|---|---|---|
| CLIP-ViT-B/16 | 11.82min | 68.34 | 0 |
| TPT | 447min | 68.98 | +0.64 |
| DiffTPT | 1346min | 70.30 | +1.96 |
| TDA | 22min | 69.51 | +1.17 |
| Dota (Ours) | 22min | 70.68 | +2.34 |

Table 4: Comparisons of our Dota with other methods in terms of efficiency (*Testing Time*) and effectiveness (*Accuracy*). The final column shows the accuracy gain compared with the baseline.

**Inference time comparison.** To illustrate the efficiency of the proposed method, we conduct evaluation about the inference time using the ViT-B/16 backbone on the ImageNet (Deng et al., 2009) dataset. The experimental results are shown in Tab. 4. From the table, we can see that the proposed method is faster than the methods that require gradient backpropagation. For example, Dota is 24 times

faster than TPT, and 61 times faster than DiffTPT. Therefore, test-time adaptation methods that require gradient backpropagation may not be applicable during deployment due to the performance limitations of the inference device. At the same time, compared with TDA, the speed of the proposed method is comparable, but the performance is higher.

## 4.2 ABLATION STUDIES AND FURTHER ANALYSIS

fa**Hyperparameters analysis.** To validate the sensitivity of our model to hyperparameters, we conduct systematic experiments and analyses. First, we evaluate the hyperparameter $\sigma^2$ while keeping other parameters fixed. The results showed minimal impact on model accuracy, with performance ranging from 70.36 to 70.68. Next, we test different $\rho$ and $\eta$ combinations, observing stable performance across combinations. For instance, accuracy ranged from 70.68 to 69.91 as $\rho$ and $\eta$ varied. Notably, all hyperparameter combinations show that the proposed method outperforms the original zero-shot classifier, indicating that TTA can significantly enhance performance even without a validation set for hyperparameter tuning.

| $\sigma^2$ | 0.0001 | 0.001 | 0.002 | 0.004 | 0.008 | 0.02 |
|---|---|---|---|---|---|---|
| **Acc** | 70.58 | 70.63 | 70.68 | 70.64 | 70.56 | 70.36 |

| $\eta\backslash\rho$ | 0.005 | 0.01 | 0.02 | 0.03 |
|---|---|---|---|---|
| 0.2 | 70.68 | 70.66 | 70.51 | 70.43 |
| 0.3 | 70.66 | 70.51 | 70.28 | 70.16 |
| 0.4 | 70.66 | 70.48 | 70.19 | 70.03 |
| 0.5 | 70.66 | 70.44 | 70.08 | 69.91 |

Table 5: Hyperparameters analysis on the $\sigma^2$ and $(\rho, \eta)$ combinations.

**The necessity of adaptive fusion of zero-shot and test-time classifier**. We conduct ablation study to show that adaptive fusion of zero-shot and test-time classifier is necessary. The specific experimental results are shown on the Tab. 5. It can be observed that as $\rho$ increases (indicating the diminishing effect of dynamic fusion), the performance of Dota consistently decreases.

**Performance analysis with limited human feedback.** We conducted experiments with human feedback ratios of 1% and 2%. The results, shown in Tab. 6, demonstrate that the model achieves absolute performance improvement even with only 1% feedback.

| Feedback ratio | Acc | Acc$^\star$ |
|---|---|---|
| 0% | 70.68 | 70.68 |
| 1% | 70.77 | 71.52 |
| 2% | 70.79 | 72.26 |

Table 6: ACC with limited feedback.

**Performance on non-i.i.d. data streams.** During testing, the distribution of test data may change continuously (Gong et al., 2022; Yuan et al., 2023; Marsden et al., 2024). To evaluate the model's robustness under test distribution shift, we conducted corresponding experiments on multiple distribution shift settings. The experimental results are shown in Tab. 7. Details are shown in Appendix A.1. We can see that the proposed method is relatively robust to the test-time distribution shift.

| **Distribution** | I.I.D | (5,0.1) | (5, 0.5) | (5,1) | (10,0.1) | (10, 0.5) | (10,1) |
|---|---|---|---|---|---|---|---|
| **Performance** | 70.68 | 70.45 | 70.52 | 70.83 | 70.39 | 70.55 | 70.66 |

Table 7: Performance comparison on i.i.d and non-i.i.d. test dataset.

| Feedback instances | 0 | 1 | 2 - 6 | 7 - 11 | 12 - 16 |
|---|---|---|---|---|---|
| Number of Classes | 300 | 203 | 390 | 92 | 15 |

Table 8: Number of feedback instances received by different classes under a 5% feedback rate.

**Analysis of continuous learning ability and test-time forgetting of TDA.** When testing on the ImageNet dataset, we record the performance of the most recent 5,000 test samples and compare them with the original zero-shot classifier performance, recording the relationship between the improvement in model performance and the number of test samples seen. The results are shown in Fig. 3. From the experimental results, we can see that the proposed method gradually improves the model performance as the number of test samples increases. In contrast, the improvement of TDA first increases and then decreases, and it is unable to continuously learn from the test data stream due to the test-time forgetting problem. We show the performance of the last 50% of test samples and all samples on more datasets in Tab. 9. The experimental results clearly show that the performance of the last 50% of test samples is significantly higher than the overall performance. The above improvement is due to the fact that the estimated distribution becomes more reliable as the number of observed test samples increases. However, TDA is different, and its performance has declined on multiple datasets.

| Method | Aircraft | Caltech101 | Cars | DTD | Flower102 | Food101 | Pets | SUN397 | UCF101 |
|---|---|---|---|---|---|---|---|---|---|
| TDA (all test samples) | 23.91 | 94.24 | 67.28 | 47.40 | 71.42 | 86.14 | 88.63 | 67.62 | 70.66 |
| TDA (last 50% test samples) | 26.57 | 93.59 | 66.95 | 46.22 | 71.75 | 86.02 | 89.26 | 67.86 | 72.20 |
| Dota (All test samples) | 25.59 | 94.32 | 69.48 | 47.87 | 74.67 | 87.02 | 91.69 | 69.70 | 72.06 |
| Dota (last 50% test samples) | 27.11 | 94.65 | 69.88 | 50.95 | 75.89 | 87.10 | 93.02 | 70.67 | 73.20 |

Table 9: Performance of `Dota` and TDA with ViT-B/16 across multiple datasets, comparing overall accuracy and the last 50% of test samples to show continuous adaptability.

**Accuracy analysis of the selected uncertain samples.** We evaluate the zero-shot classification accuracy of the selected uncertain samples. The experimental results are shown in Tab. 10. From the table, we can see that the uncertain samples found using the proposed confidence-based method usually have lower zero-shot classification accuracy. The zero-shot classifier averages 64.59% accuracy, but for the 5% uncertain samples found by our method, it drops to 25.87%. This demonstrates that the proposed method accurately detects samples with low classification confidence, enabling efficient label collection through a human-in-the-loop approach. Confidence is more effective than similarity in identifying uncertain samples, as it accounts for similarities across multiple classes, while maximum similarity focuses on just one class.

| Feedback Percentile | Method | Aircraft | Caltech101 | Cars | DTD | EuroSAT | Flower102 | Food101 | Pets | SUN397 | UCF101 | Average |
|---|---|---|---|---|---|---|---|---|---|---|---|---|
| - | CLIP | 23.22 | 93.55 | 66.11 | 45.04 | 50.42 | 66.99 | 82.86 | 86.92 | 65.63 | 65.16 | 64.59 |
| 5% | Random | 19.17 | 84.09 | 51.14 | 47.54 | 37.06 | 71.59 | 76.2 | 93.33 | 67.72 | 62.62 | 61.05 |
| | Similarity | 19.32 | 91.95 | 51.76 | 30.36 | 5.00 | 42.86 | 54.56 | 50.00 | 55.61 | 32.22 | 43.36 |
| | Confidence | 11.80 | 68.35 | 25.00 | 15.87 | 20.51 | 9.63 | 31.74 | 37.93 | 19.79 | 18.09 | 25.87 |
| 15% | Random | 17.16 | 84.81 | 58.9 | 44.72 | 38.53 | 65.67 | 77.5 | 89.72 | 63.71 | 58.94 | 59.97 |
| | Similarity | 21.37 | 95.74 | 58.94 | 32.70 | 13.73 | 37.89 | 63.40 | 65.74 | 55.91 | 45.68 | 49.11 |
| | Confidence | 11.36 | 71.81 | 29.81 | 18.12 | 19.63 | 20.16 | 44.91 | 52.04 | 30.66 | 21.42 | 31.99 |

Table 10: Top-1 accuracy (%) of uncertainty samples selected by different methods. Lower accuracy suggests better identification of uncertain samples by the method.

**Analysis of uncertain samples.** To illustrate test-time adaptation with human feedback, we analyze the distribution of feedback across ImageNet classes under a 5% feedback rate. From the Tab. 8, it can be seen that the amount of human feedback collected for different categories is imbalanced, with some categories receiving more feedback and others receiving less. Similar conclusions were also observed in active learning of CLIP (Bang et al., 2024). These findings highlight the potential for further refinement of the methods. Addressing the observed class imbalance during sample selection and human feedback acquisition could further enhance the effectiveness of our approach.

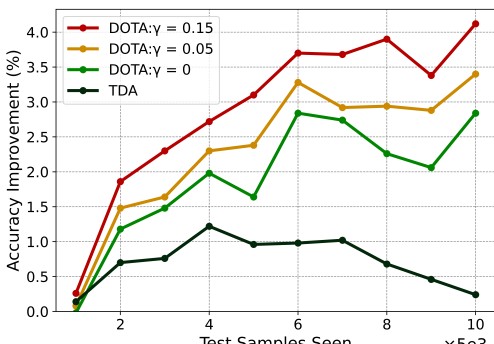

Figure 3: Improvement of different methods in model performance as the number of encountered test samples increases.

## 5 CONCLUSION AND FUTURE WORK

We propose a method for continuous test-time adaptation, which enhances the original zero-shot classifier by continually adapting through online estimation of the test sample distribution and obtaining test-time posterior probabilities. To achieve this, we introduce an online distribution parameter estimation method that can estimate the distribution of test samples during testing by using the prediction probabilities from the zero-shot classification of the data stream samples. Additionally, to further adapt to uncertain samples that the base model may encounter during deployment, this work is the first to define the task of test-time adaptation, which detects uncertain samples and collects human feedback labels. By leveraging the human feedback on uncertain samples, the proposed continuous adaptation method is further improved. `Dota` demonstrates superior performance and comparable speed across various scenarios. In the future, we believe that exploring better test-time uncertainty estimation methods to collect human feedback and conduct test-time adaptation represents a promising direction in Human-AI collaboration.

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

# A APPENDIX

## A.1 RESULTS ACROSS MULTIPLE DATA ORDERING.

We conducted experiments on five test data in different ordering on multiple datasets. The experimental results are shown in the Tab. 11. From the experimental results, we can see that the order of test data has little effect on the prediction performance.

| Dataset | Method\Data ordering | 1 | 2 | 3 | 4 | 5 | Average |
|---------|---------------------|-----|-----|-----|-----|-----|---------|
| ImageNet | Dota | 70.57 | 70.70 | 70.70 | 70.61 | 70.45 | 70.61 |
| ImageNet | Dota with 5% human feedback | 71.09 | 71.10 | 71.00 | 70.85 | 70.90 | 70.99 |
| ImageNet | Dota with 15% human feedback | 71.86 | 71.76 | 71.74 | 71.85 | 71.93 | 71.83 |
| eurosat | Dota | 57.78 | 56.99 | 58.15 | 57.28 | 57.22 | 57.48 |
| eurosat | Dota with 5% human feedback | 64.89 | 65.38 | 64.74 | 64.60 | 66.28 | 65.18 |
| eurosat | Dota with 15% human feedback | 75.68 | 77.90 | 76.85 | 77.57 | 76.93 | 76.99 |
| OxfordPets | Dota | 91.88 | 91.71 | 91.80 | 91.71 | 91.99 | 91.82 |
| OxfordPets | Dota with 5% human feedback | 92.56 | 93.05 | 92.78 | 92.89 | 92.56 | 92.77 |
| OxfordPets | Dota with 15% human feedback | 93.84 | 93.68 | 93.68 | 93.62 | 93.70 | 93.70 |

Table 11: Experimental results across multiple datasets and test data orderings.

We conducted additional experiments to evaluate the model's performance under non-i.i.d. data distribution during testing, using the ImageNet dataset as a benchmark. By employing a Dirichlet distribution, we simulated varying degrees of non-i.i.d. data streams, adjusting the concentration parameter and dividing the dataset into 5 and 10 time slices for analysis. The details of the experiments are shown as follows:

**Time Slices**: We divided the ImageNet dataset into 5 and 10 time slices, where each slice contains varying numbers of samples and class distributions.

**Concentration Parameter** ($[\alpha]_K$): The concentration parameter of the Dirichlet distribution controls the uniformity of class distributions across slices. Smaller $\alpha$ values (e.g., 0.1) create highly uneven distributions, while larger values (e.g., 0.5 and 1) result in more uniform distributions.

**Evaluation Setting**: Since the sizes of sub-datasets for each time slice are unequal, the final average accuracy is a weighted average based on the number of samples in each slice. The experimental results are summarized below.

| $\alpha$ | Slice 1 | Slice 2 | Slice 3 | Slice 4 | Slice 5 | Average |
|------|---------|---------|---------|---------|---------|---------|
| 0.1 | 68.67 | 69.72 | 71.60 | 71.14 | 71.08 | 70.45 |
| 0.5 | 69.57 | 70.75 | 69.53 | 71.85 | 70.83 | 70.52 |
| 1 | 69.47 | 71.59 | 71.06 | 69.92 | 71.83 | 70.83 |

Table 12: Performance on non-i.i.d. data streams (5 slices).

| $\alpha$ | Slice 1 | Slice 2 | Slice 3 | Slice 4 | Slice 5 | Slice 6 | Slice 7 | Slice 8 | Slice 9 | Slice 10 | Average |
|------|---------|---------|---------|---------|---------|---------|---------|---------|---------|----------|---------|
| 0.1 | 70.44 | 69.48 | 67.26 | 70.25 | 71.49 | 68.91 | 73.03 | 70.48 | 69.72 | 72.39 | 70.39 |
| 0.5 | 69.01 | 69.30 | 72.10 | 71.04 | 69.83 | 69.90 | 70.30 | 70.63 | 72.34 | 70.81 | 70.55 |
| 1 | 67.23 | 70.12 | 71.01 | 68.25 | 69.76 | 71.14 | 72.01 | 72.59 | 71.46 | 71.75 | 70.66 |

Table 13: Performance on non-i.i.d. data streams (10 slices).

From the experimental results, we can see that the model shows strong robustness to non-i.i.d. data streams, with only minimal accuracy decline under small $\alpha$ (e.g., $\alpha = 0.1$). Moreover, for relatively mild test distribution changes, our approach adapts well by incorporating human feedback and online distribution estimation. Simple modifications, such as a sliding window mechanism, could further improve performance. However, in extreme distribution shift scenarios, performance may be impacted due to challenges in reliably estimating the distribution with insufficient samples.

## A.2 MORE DETAILS AND EXPLANATION ABOUT THE $f_k(x)$ IN EQ. 2.

The function $f_k(\boldsymbol{x})$, often referred to as the *discriminant function*, measures how well a data point $\boldsymbol{x}$ fits the distribution of class $k$. It is derived from Gaussian Discriminant Analysis and consists of two main components. The first component is the Mahalanobis distance, $-\frac{1}{2}(\boldsymbol{x} - \mu_k)^T \Sigma_k^{-1} (\boldsymbol{x} - \mu_k)$, which calculates the squared distance between $\boldsymbol{x}$ and the class mean $\mu_k$, scaled by the inverse of the covariance matrix $\Sigma_k$. This term captures the similarity of $\boldsymbol{x}$ to the center of the class, considering

feature correlations. The second component is the normalization term, $-\frac{1}{2}\log|\Sigma_k|$, which accounts for the determinant of the covariance matrix $\Sigma_k$ and reflects the spread (or volume) of the Gaussian distribution for class $k$. This ensures that classes with larger variances are normalized appropriately. Intuitively, a larger value of $f_k(\boldsymbol{x})$ indicates a higher likelihood that $\boldsymbol{x}$ belongs to class $k$. In classification, $f_k(\boldsymbol{x})$ is used within the softmax function to compute the posterior probability $P(y=k\mid\boldsymbol{x})$, which determines the most likely class for $\boldsymbol{x}$: $P(y=k\mid\boldsymbol{x})=\frac{\exp(f_k(\boldsymbol{x}))}{\sum_{k=1}^{K}\exp(f_k(\boldsymbol{x}))}$.

## A.3   PERFORMANCE COMPARISON BETWEEN DIFFERENT SAMPLE SELECTION METHOD

We incorporate more sample selection strategies. The experimental results are shown in the table below. From the experimental results in Tab. A.3, it can be seen that the confidence-based selection can achieve better performance.

| Human-Feedback | Method | Acc | Acc$^\star$ |
|---|---|---|---|
| 0% | Dota | 70.68 | 70.68 |
| 5% | Random | 70.86 | 72.34 |
| | Similarity | 71.08 | 73.48 |
| | Confidence | 71.01 | 74.52 |
| 15% | Random | 71.28 | 75.61 |
| | Similarity | 71.68 | 78.18 |
| | Confidence | 71.83 | 80.91 |

Table 14: Experimental results with different human-feedback percentages and selection strategies.

---

**Algorithm 1:** The distributional test-time adaptation pseudocode of `Dota`.

---

**Input:** The embedding of $N$ test samples $\{\boldsymbol{x}_n\}_{n=1}^{N}$, zero-shot classification weights
  $[\boldsymbol{w}_1,\cdots,\boldsymbol{w}_K]$;
Initializing the distribution of different class;
**for** *each test sample $\boldsymbol{x}_i$* **do**
  Obtain the zero-shot classification probability with Eq. 1;
  Determine whether $\boldsymbol{x}_i$ is an uncertain sample according to Sec. 3.2;
  Collect human feedback if needed;
  Update the distribution of different class with Eq. 4;
  Obtain the test-time classification probability with Eq. 2;
  Obtain the final classification result with Eq. 5.

---

## A.4   DETAILS OF COMPARISON METHOD.

We compare the proposed method with the following method: (1) TPT (Shu et al., 2022) is a test time prompt tuning method. (2) DiffTPT (Feng et al., 2023) introduces more diverse test sample augmentation with diffusion model. TPT and DiffTPT require gradient backpropagation to update prompt, so they require greater computational cost. (3) TDA (Karmanov et al., 2024) introduce an efficient test-time adaption method do not need backpropagation, which works with a cache containing representative samples to conduct test time adaption with these samples. To be consistent with the previous works (Shu et al., 2022; Karmanov et al., 2024), we also include the baseline zero-shot performance of CLIP, using the ensemble of 80 hand-crafted prompts (Radford et al., 2021).

## A.5   THE NECESSITY OF ESTIMATING DISTRIBUTION WITH ZERO-SHOT PROBABILITY.

We compared the performance of the `Dota` with a simplified version that only uses high-confidence samples to estimating the distribution of different calsses. This experiment aimed to understand the necessity of estimating distribution with zero-shot probability rather than high-confidence samples. The experimental results are shown in Tab. A.5. From the experimental results, we can see that in most cases, using all data to update the distribution parameters will not lead to a decrease in model

performance, but will help improve the performance of the model. These findings highlight the importance of low confidence samples.

| Method | Caltech101 | Cars | DTD | EuroSAT | Flower102 | Food101 | Pets | SUN397 | UCF101 | Average |
|---|---|---|---|---|---|---|---|---|---|---|
| TDA | 94.24 | 67.28 | 47.40 | 58.00 | 71.42 | 86.14 | 88.63 | 67.62 | 70.66 | 72.38 |
| Dota (learn from high confidence samples) | 94.10 | 68.08 | 45.70 | 58.60 | 72.06 | 88.06 | 89.47 | 69.90 | 68.80 | 72.75 |
| Dota | 94.32 | 69.48 | 47.87 | 57.65 | 74.67 | 87.02 | 91.69 | 69.70 | 72.06 | 73.49 |

Table 15: Ablation study comparing the performance of `Dota` with a variant that uses only the high confidence samples to estimate the distribution parameters.

## A.6 EFFECTIVENESS OF TTA WITH HUMAN FEEDBACK ON LARGE-SCALE TEST DATASET.

We evaluate the impact of incorporating human feedback on model performance using a larger dataset (over 1 million test samples). Specifically, we introduce more human feedback during the early stages of model testing (the first 50,000 samples), but stop introducing feedback or updating the model in the later stages of testing. The experimental results are as follows. It can be seen that when the model is adapted during testing with human feedback, the more test samples the model has in the future, the greater the benefits it brings, and the lower the cost of human feedback.

| Model | Performance (%) in terms of standard ACC |
|---|---|
| Original CLIP | 70.14 |
| DOTA without human feedback | 70.89 |
| DOTA with Feedback rate at 0.75% | 72.01 |
| DOTA with Feedback rate at 1% | 72.44 |
| DOTA with Feedback rate at 2% | 73.15 |

Table 16: Experimental results with different human-feedback percentages on large-scale dataset.

## A.7 THE NECESSITY OF DISTRIBUTION ESTIMATION.

We compared the performance of the `Dota` with a simplified version that only uses the mean, excluding the estimation of the Gaussian distribution by removing the covariance matrixs. This experiment aimed to understand the necessity of continual distribution estimation in enhancing model accuracy. The experimental results are shown in Tab. 17. The third row in the table presents the accuracy reductions across different datasets when the covariance matrix is removed. The results indicate a consistent decrease in accuracy across all datasets, with a particularly notable drop of 3.41% on the UCF101 dataset. These findings highlight the importance of continual distribution estimation.

| Method | Aircraft | Caltech101 | Cars | DTD | EuroSAT | Flower102 | Food101 | Pets | SUN397 | UCF101 | Average |
|---|---|---|---|---|---|---|---|---|---|---|---|
| Dota | 25.59 | 94.32 | 69.48 | 47.87 | 57.65 | 74.67 | 87.02 | 91.69 | 69.70 | 72.06 | 69.01 |
| w/o covariance | 24.99 | 92.09 | 67.29 | 45.62 | 54.99 | 70.89 | 86.40 | 90.11 | 67.62 | 68.65 | 66.87 |
| | -0.60 | -2.23 | -2.19 | -2.25 | -2.66 | -3.78 | -0.62 | -1.58 | -2.08 | -3.41 | -2.14 |

Table 17: Ablation study comparing the performance of `Dota` with a variant that uses only the mean, excluding the estimation of the Gaussian distribution (by removing the covariance matrix). The significant drop (third row) in model performance without distribution estimation highlights the importance of distributional test-time adaptation.

## A.8 EFFECTS OF DIFFERENT UNCERTAINTY SAMPLE SELECTION STRATEGIES.

To evaluate the effectiveness of the proposed confidence-based test-time uncertainty estimation for selecting samples to collect human feedback , we designed two alternative strategies for comparison. First, we randomly selected inference samples for human feedback. Second, we replaced the confidence in the proposed method (as described in Sec. 3.2) with the maximum cosine similarity. The experimental results, shown in Tab. 18, demonstrate that the confidence-based uncertainty sample selection method significantly improves test-time adaptation performance compared to random selection and the cosine similarity-based approach. However, designing more effective methods for

identifying uncertain samples to collect human feedback remains an open problem, which we leave for future exploration.

| Feedback Percentile | Method | Aircraft | Caltech101 | Cars | DTD | EuroSAT | Flower102 | Food101 | Pets | SUN397 | UCF101 | Average |
|---|---|---|---|---|---|---|---|---|---|---|---|---|
| 5% | Random | 26.58 | 94.36 | 70.22 | 48.94 | 65.25 | 75.48 | 87.08 | 92.07 | 70.18 | 73.43 | 70.36 |
| | Similarity | 27.06 | 94.36 | 70.30 | 50.24 | 63.38 | 76.17 | 87.11 | 92.42 | 70.28 | 74.41 | 70.57 |
| | Confidence | 26.73 | 94.56 | 70.95 | 49.82 | 65.00 | 76.86 | 87.17 | 92.78 | 70.49 | 75.26 | 70.96 |
| 15% | Random | 28.68 | 94.69 | 71.57 | 50.83 | 74.63 | 76.37 | 87.15 | 92.34 | 70.93 | 75.71 | 72.29 |
| | Similarity | 29.46 | 94.56 | 72.27 | 53.84 | 71.09 | 78.97 | 87.24 | 93.08 | 71.42 | 76.55 | 72.85 |
| | Confidence | 28.65 | 95.01 | 73.01 | 53.78 | 76.60 | 79.70 | 87.41 | 93.54 | 71.82 | 79.33 | 73.89 |

Table 18: Top-1 accuracy (%) of experimental results using the ViT-B/16 backbone with different methods for selecting uncertainty samples for human feedback. Random, Similarity, and Confidence refer to Randomly selecting inference samples, selecting based on zero-shot cosine similarity, and selecting based on the confidence of the zero-shot classifier, respectively.

## A.9 Implementation details.

All the models in our experiments are built upon the pre-trained CLIP model (Radford et al., 2021) that consists of an image encoder and a text encoder. Test-time adaptation is set for single-image scenarios, using a batch size of 1. For natural distribution shifts scenario, we tune all our hyperparameters using the single ImageNet validation set. For the cross-domain generalization scenario, we perform hyperparameter search using the corresponding validation sets. We adjust $\sigma^2$ within [0.001, 0.002, 0.004], then search for the best $\eta$ across [0.2, 0.3, 0.4, 0.5] and $\rho$ across [0.005, 0.01, 0.02, 0.03], with the shrinkage parameter $\epsilon$ set to 0.0001. We use top-1 accuracy (%) as our evaluation metric. All experiments are conducted using a single NVIDIA RTX 4090 GPU and a 12-core Intel Xeon Platinum 8352V CPU.

## A.10 Limitations and future works.

Here we briefly discuss the limitations of our method and outline potential directions for future work. (1) While our approach demonstrates the advantage of continuously estimating the distribution of test data, allowing for adaptation to test data, it does not consistently outperform TDA on all the dataset. For example, as shown in Tab. 19, on ImagenetV2 (Recht et al., 2019) datasets with only 10 samples per class, Dota does not significantly exceed TDA. However,

| Method | ViT-B/16 | ResNet-50 |
|---|---|---|
| CLIP | 61.88 | 52.91 |
| TDA | 64.67 | 55.54 |
| Dota (All test samples) | 64.41 | 55.27 |
| Dota (The last 50% of test samples ) | 65.06 | 55.82 |

Table 19: Comparisons of our Dota with other methods on the ImageNetV2 dataset, where each class contains only 10 samples.

its performance on the last 50% of the test samples shows a clear improvement. This indicates that the proposed model has the potential to further improve as more test samples becomes available. Moreover, as demonstrated in Fig. 3, our method gradually outperforms TDA over time. To avoid the limitation, a promising way for future research is designing a mechanism to evaluate the reliability of the adapter, allowing dynamic decisions on whether to introduce it based on its reliability. (2) This paper also introduces the novel task of test-time adaptation with human feedback and proposes an initial approach. Future work could focus on refining methods to accurately detect unreliable samples and selectively incorporate human feedback, providing a valuable direction for further improvement.

**Broader impact.** Foundational models are being widely deployed, but they do not always adapt perfectly to the distribution of test data. Collecting new data and fine-tuning models for specific applications can be costly and slow in response. Therefore, allowing models to adapt to unseen data during test time can enhance their generalization and adaptability. This approach has potential in fields like healthcare and assistive technologies, as it can help reduce subgroup bias caused by insufficient data for minority groups during training and improve fairness.

| Dataset | Classes | Validation Size | Test Size | Task |
|---------|---------|-----------------|-----------|------|
| ImageNet | 1,000 | N/A | 50,000 | Classification |
| ImageNet-V2 | 1,000 | N/A | 10,000 | Generalization |
| ImageNet-S | 1,000 | N/A | 50,889 | Generalization |
| ImageNet-A | 200 | N/A | 6,862 | Generalization |
| ImageNet-R | 200 | N/A | 30,000 | Generalization |
| Aircraft | 100 | 3,333 | 3,333 | Aircraft recognition |
| Caltech101 | 100 | 1,649 | 2,465 | Object recognition |
| Cars | 196 | 1,635 | 8,041 | Car recognition |
| DTD | 47 | 1,128 | 1,692 | Texture classification |
| EuroSAT | 10 | 5,400 | 8,100 | Remote sensing classification |
| Flowers102 | 102 | 1,633 | 2,463 | Flower recognition |
| Food101 | 101 | 20,200 | 30,300 | Food classification |
| Pets | 37 | 736 | 3,669 | Pet classification |
| SUN397 | 397 | 3,970 | 19,850 | Scene recognition |
| UCF101 | 101 | 1,898 | 3,783 | Action recognition |

Table 20: Datasets details.

