# OpenReview forum: "DOTA: Distributional Test-Time Adaptation of Vision-Language Models"
_ICLR.cc/2025/Conference — Submitted to ICLR 2025_

### Official Review · Reviewer_GKJd · 2024-10-18

**Soundness:** 3
**Presentation:** 3
**Contribution:** 2
**Rating:** 6
**Confidence:** 4

**Summary:**

The authors propose a Distributional Test-time Adaptation (DOTA) method, which adapt the pretrained Vision-language foundation models (e.g., CLIP) to the test target domain by estimating the distributions of different categories for test samples continually. The authors further introduce a human feedback collaboration method which identifies uncertain samples to further enhance the adaptability. Extensive experiments on diverse datasets validate the effectiveness of the proposed method.

**Strengths:**

1. The writing and figures are good and easy to understand.

2. The DistributiOnal Test-time Adaptation (DOTA) method for Vision-language foundation models without BP is simple yet effective during testing in new target domain, achieving a significant improvement compared to current state-of-the-art methods in most of the datasets.

3. This paper first define the test-time adaptation problem with human feedback, allows the test-time adaptation for uncertain samples with human feedback.

4. An adaptive final fusion probability is introduced to mitigate the potential negative impact when the number of test samples is insufficient.

**Weaknesses:**

1. The proposed method estimate the data distribution of samples in the current test environment during testing, but there lack some evidence or visualization. Why updating the feature distribution for different categories works better?

2. The DOTA method seems somewhat similar to the T3A method [R1], which continually maintains a memory bank for prototypes during the testing stage. Could the authors clarify and analysis the difference and the advantages of the proposed method?

[R1] Test-Time Classifier Adjustment Module for Model-Agnostic Domain Generalization

3. The proposed method seems works for not only VLMs but also other models in all classification tasks. Is it suitable for traditional TTA or Domain Generalization tasks?

4. There could give more details and explanation about the $f_k(x)$ in Eq.(3).

5. The uncertainty estimation method is simple. Is there any other uncertainty estimation method (like entropy) better?

6. There missing an ablation study for the adaptive fusion probability.

**Questions:**

see the Weaknesses

---

> ### Author Response · Authors · 2024-11-16
> **Official Response by Authors (1/4)**
>
> ## Weakness 1. Why updating the feature distribution for different categories works better?
>
> The proposed online data estimation method is based on the method in the existing paper[1]. Theoretically, for any sequence, the average regret of the proposed method converges to zero in the limit. This means that the proposed method can finally obtain an accurate estimation of the test data distribution, leading to better classification performance.
>
> In addition, we visualized the improvement in accuracy during the test process, and we can see that as the number of test samples increases, the model performance continues to improve.
>
> | Test Samples Seen (5e^3) | 1    | 2    | 3    | 4    | 5    | 6    | 7    | 8    | 9    | 10   |
> |--------------------------|------|------|------|------|------|------|------|------|------|------|
> | DOTA Improvements         | 0.02 | 1.18 | 1.48 | 1.98 | 1.64 | 2.84 | 2.74 | 2.26 | 2.06 | 2.84 |
>
>
> ## Weakness 2. Difference and the advantages of the proposed method compared with T3A.
>
> T3A is similar to TDA [2], an important baseline and reference in this paper, as both store representative test samples to guide the classification of subsequent ones. However, this approach represents an inefficient use of test samples. Specifically, T3A and TDA naively maintain a limited cache of representative samples during testing, dynamically updating the cache with higher-confidence samples. This strategy results in test-time forgetting, as older cached samples are discarded when new confident samples are added.
>
> In contrast, our proposed method does not rely on preserving representative features. Instead, it continuously estimates the distribution of test samples, allowing the model to adapt dynamically to the test environment. This design enables the model to maintain long-term adaptability to changes in the environment, avoiding the pitfalls of test-time forgetting. The related claims are also experimentally verified in Figure 3.
>
> ## Weakness 3. Is it suitable for traditional TTA or Domain Generalization tasks?
>
> The proposed method is a general continuous estimation method. However, in non-visual language models, appropriate adjustments need to be made to adapt to changes in application scenarios. For example, the initialization of the mean feature vectors of different categories needs to be adjusted. Due to the limitations of the length and focus of the article, we will leave the extension to other scenarios for future work.
>
> The proposed method is also applicable to Domain Generalization tasks. We report the results of some Domain Generalization tasks datasets (variant datasets of ImageNet) in the paper, which shows that the proposed method can achieve good performance.

---

> ### Author Response · Authors · 2024-11-16
> **Official Response by Authors (2/4)**
>
> ## Weakness 4. More details and explanation about the $f_k(x)$ in Eq.(3).
>
> Thank you for your suggestion. We will include the corresponding explanation in future revisions. Equation 3 is derived from the classic Gaussian Discriminant Analysis algorithm, which constructs the classifier by estimating the data distribution for different categories. A detailed derivation process and explanation will be added in future versions to ensure clarity. Here we offer a shortened explanation.
>
> ### Explanation of $f_k(\mathbf{x})$:
>
> The function $f_k(\mathbf{x})$, often referred to as the **discriminant function**, measures how well a data point $\mathbf{x}$ fits the distribution of class $k$. It combines two key components of Gaussian Discriminant Analysis:
>
> 1. **Mahalanobis Distance**:
>    $$
>    -\frac{1}{2} (\mathbf{x} - \mu_k)^T \Sigma_k^{-1} (\mathbf{x} - \mu_k)
>    $$
>    This term calculates the squared Mahalanobis distance between $\mathbf{x}$ and the class mean $\mu_k$, scaled by the covariance matrix $\Sigma_k$. It measures how far $\mathbf{x}$ is from the mean of class $k$, taking into account the variance and correlation of features within that class.
>
> 2. **Normalization Term**:
>    $$
>    -\frac{1}{2} \log |\Sigma_k|
>    $$
>    This term accounts for the determinant of the covariance matrix $\Sigma_k$, which represents the spread (volume) of the Gaussian distribution for class $k$. It ensures that classes with larger variances are appropriately normalized.
>
> ### Intuition:
> - A larger $f_k(\mathbf{x})$ value indicates a higher likelihood that $\mathbf{x}$ belongs to class $k$.
> - $f_k(\mathbf{x})$ combines both the distance from the mean (how similar $\mathbf{x}$ is to the class center) and the shape of the distribution (to adjust for variances).
>
> In the context of classification, $f_k(\mathbf{x})$ is used in the softmax function to compute the posterior probability $P(y = k | \mathbf{x})$, which determines the most likely class for $\mathbf{x}$:
> $$
> P(y = k | \mathbf{x}) = \frac{\exp(f_k(\mathbf{x}))}{\sum_{k=1}^K \exp(f_k(\mathbf{x}))}.
> $$

---

> ### Author Response · Authors · 2024-11-16
> **Official Response by Authors (3/4)**
>
> ## Weakness 5. The uncertainty estimation method is simple.
>
> Thank you for your suggestion regarding uncertain sample selection. Based on your feedback, we extended our experiments to include entropy and variance-based methods in addition to our initial confidence-based baseline. We evaluated these methods on multiple datasets and tested their accuracy on the selected uncertain samples. The results are summarized below:
>
> #### Experimental Results:
>
> | Dataset         | **Acc of All Samples** | **Variance (5%)** | **Entropy (5%)** | **Softmax (5%)** | **Variance (15%)** | **Entropy (15%)** | **Softmax (15%)** |
> |-----------------|-------------------------|-------------------|------------------|------------------|--------------------|-------------------|-------------------|
> | **ImageNet**    | 68.78                  | 22.49            | 28.16           | 19.85           | 29.87             | 33.89            | 28.84            |
> | **ImageNet-A**  | 59.94                  | 21.25            | 21.45           | 21.56           | 27.24             | 28.78            | 27.07            |
> | **ImageNet-V2** | 64.17                  | 18.51            | 20.70           | 17.63           | 24.94             | 26.85            | 25.50            |
> | **ImageNet-R**  | 80.44                  | 22.49            | 23.69           | 21.23           | 36.10             | 36.98            | 35.96            |
> | **ImageNet-S**  | 49.42                  | 6.76             | 8.05            | 6.59            | 11.98             | 13.31            | 11.82            |
>
> ---
>
> From the experimental results, the confidence-based method has the best uncertainty sample selection ability. The selected samples are more likely to be misclassified by the model.
>
> To strengthen our evaluation, we added a more competitive baseline for test-time adaptation (TTA) based on human feedback. Since we are the first to define this specific problem, there are few directly related methods available for comparison. To address this, we compared our approach to a concurrent submission (ATPT [3]) to ICLR.
>
> The comparison results, using 5% human feedback, are summarized in the table below. The proposed method significantly outperforms ATPT.
>
> | Model                  | Average | Aircraft | Caltech101 | Cars  | DTD   | EuroSAT | Flower102 | Food101 | Pets  | SUN397 | UCF101 |
> |------------------------|---------|----------|------------|-------|-------|---------|-----------|---------|-------|--------|--------|
> | **ATPT (5% feedback)** | 67.26   | 24.85    | 94.27      | 67.86 | 48.23 | 49.88   | 72.36     | 86.77   | 90.65 | 67.51  | 70.23  |
> | **Ours (5% feedback)** | 70.96   | 26.73    | 94.56      | 70.95 | 49.82 | 65.00   | 76.86     | 87.17   | 92.78 | 70.49  | 75.26  |
>
>
>
> Moreover, we are also changing the previous comparison method to introduce human feedback and provide a stronger baseline. Specifically, we added an uncertain sample cache on the TDA to maintain the mean of uncertain samples. The experimental results are shown below. The experimental results indicate that the proposed method achieves superior performance. However, TDA may fail in certain cases due to its inefficient utilization of human feedback samples. Specifically, merely storing uncertain samples and their labels in the cache does not lead to performance improvement, possibly because the features of uncertain samples may contain data noise.
>
>
> | Method                | ImageNet | Average except ImageNet | FGVC  | Caltech101 | Cars  | DTD   | EuroSAT | Flower | Food101 | Pets  | SUN397 | UCF101 |
> |-----------------------|----------|---------|-------|------------|-------|-------|---------|--------|---------|-------|--------|--------|
> |TDA 5% Human Feedback     | 69.46    | 65.31   | 23.13 | 91.36      | 64.73 | 41.78 | 55.54   | 69.47  | 85.87   | 89.48 | 64.54  | 67.17  |
> | Ours with 5% feedback | 71.01| 70.96   | 26.73    | 94.56      | 70.95 | 49.82 | 65.00   | 76.86     | 87.17   | 92.78 | 70.49  | 75.26  |
> |TDA 15% Human Feedback    | 69.68    | 67.44   | 23.73 | 91.93      | 66.02 | 44.27 | 64.06   | 70.52  | 85.97   | 90.52 | 65.80  | 71.56  |
> | Ours with 15% feedback |71.83| 73.89|28.65 | 95.01 | 73.01 | 53.78 | 76.60 | 79.70 | 87.41 | 93.54 | 71.82 | 79.33 |

---

> ### Author Response · Authors · 2024-11-16
> **Official Response by Authors (4/4)**
>
> ## Weakness 6. Ablation study for adaptive fusion probability.
> We conducted corresponding experiments and analysis. The table illustrates the performance improvements achieved with different $\eta$ and $\rho$ values compared to a standard zero-shot classifier. When $\rho$ is fixed, performance decreases as $\eta$ increases, indicating that higher $\eta$ values negatively impact performance. Conversely, when $\eta$ is fixed, performance gradually declines as $\rho$ increases, reflecting a consistent downward trend. However, the model's performance remains relatively robust to changes in these hyperparameters. The experiments were carried out using a ViT-B/16 backbone on the ImageNet validation set.
>
> **It can be observed that as $\rho$ increases (indicating the diminishing effect of dynamic fusion), the performance of Dota consistently decreases.**
>
> | $\eta \backslash \rho$ | 0.005 | 0.010 | 0.020 | 0.030 |
> |-------------------------|-------|-------|-------|-------|
> | 0.2                    | 2.34  | 2.32  | 2.17  | 2.09  |
> | 0.3                    | 2.32  | 2.17  | 1.94  | 1.82  |
> | 0.4                    | 2.32  | 2.14  | 1.85  | 1.69  |
> | 0.5                    | 2.32  | 2.10  | 1.74  | 1.57  |
>
>
> [1] On-line estimation with the multivariate gaussian distribution
>
> [2] Efficient test-time adaptation of vision-language model
>
> [3] Active test time prompt learning in vision-language models

---

> > ### Comment · Reviewer_GKJd · 2024-11-25
> > **Tanks for the clarifications**
> >
> > Dear authors,
> >
> > Thank you for your detailed clarification.
> > I think the T3A method also dynamically updates a memory bank without removing any older samples, as they use the merge operation (U) for the support set. And they use the high-confidence samples for updating but the Dota method uses all samples. How do the low-confidence samples affect the performance of the proposed Dota method? For example, if some bad features are involved during the testing domain (no human feedback), will the updated distribution be robust enough compared to T3A?

---

> ### Author Response · Authors · 2024-11-25
> **Further response.**
>
> ## Comparison with T3A
>
> T3A is a classical test-time adaptation method. However, if T3A does not discard old samples, the repository grows larger as the number of test samples increases, leading to higher storage and computation costs. Specifically, its storage and computational complexity are $O(n)$, where $n$ is the number of test samples. In contrast, our method only maintains the mean and covariance matrix without requiring additional storage space. Thus, both its storage and computational complexity are $O(1)$.
>
> ## On the negative impact of low-confidence samples:
>
> 1. In fact, our initial version of the proposed method updated model parameters using only high-confidence samples, following the strategy used in T3A and TDA methods. However, the performance improvement was limited compared with TDA. The fundamental reason is that low-confidence predictions also contain information that is beneficial for model training. The experimental results are shown in the following table
>
> |              | Caltech101 | Cars  | DTD   | EuroSAT | Flower102 | Food101 | Pets  | SUN397 | UCF101 | Mean      |
> |--------------|------------|-------|-------|---------|-----------|---------|-------|--------|--------|-----------|
> | TDA         | 94.24      | 67.28 | 47.40 | 58.00   | 71.42     | 86.14   | 88.63 | 67.62  | 70.66  | 72.38     |
> | DOTA trained with high confidence samples          | 94.10      | 68.08 | 45.70 | 58.60   | 72.06     | 88.06   | 89.47 | 69.90  | 68.80  | 72.75     |
> | DOTA trained with all samples       | 94.32      | 69.48 | 47.87 | 57.65   | 74.67     | 87.02   | 91.69 | 69.70  | 69.01  | 73.49     |
>
> 2. Theoretically, the objective function of the proposed method can also be equivalent to certain prior test-time adaptation approaches, such as entropy minimization. Specifically, previous entropy minimization methods can be viewed as minimizing the cross-entropy loss between the current predicted class probability and the predicted class probability.
>
> 3. This process can also be interpreted as a single iteration of the Expectation-Maximization (EM) algorithm \citep{moon1996expectation}. Importantly, just like traditional EM algorithms, our method uses the predicted probabilities of the old classifier to train the new classifier. Specifically, obtaining the zero-shot classification probability corresponds to the expectation step, while estimating $\{{\mu_k, \Sigma_k}\}$ based on the zero-shot predicted probability corresponds to the maximization step, adhering to the principle of maximum likelihood estimation.
>
> 4. Moreover, the estimation process can also be intuitively understood as reweighting, where the zero-shot predicted probabilities (usually well calibrated[1]) are used as weights to adjust the contributions of samples to different classes, thereby mitigating the impact of the potential inaccuracies in the zero-shot predicted probabilities.
>
> 5. Finally, in practice, on ImageNet test set, each class contains only 50 samples, which may not be sufficient for the model to select a large enough number of high-confidence test samples for distributional test-time adaptation. Therefore, it is crucial to make effective use of the information from low-confidence samples. Meanwhile, due to the inherent biases of CLIP, it may naturally exhibit low confidence for certain categories, making it difficult to adapt to these categories when using only high-confidence samples.
>
> However, we also very  acknowledge your perspective that, in some cases, using only high-confidence samples can also be a good choice, and it serves as a complementary approach to the method we proposed.
>
> [1] Revisiting the calibration of modern neural networks.

---

> > ### Author Response · Authors · 2024-11-27
> >
> > Dear Reviewer, since November 27th is the final deadline for submitting the manuscript, I would like to know if you have any concerns about our manuscript that we could address through revisions to improve its score.

---

> ### Author Response · Authors · 2024-11-27
> **Further manuscript update**
>
> Dear reviewer, we have update the manuscript with our latest discussion and promise to improve it further.

---

### Official Review · Reviewer_DNWr · 2024-11-01

**Soundness:** 2
**Presentation:** 3
**Contribution:** 2
**Rating:** 6
**Confidence:** 4

**Summary:**

The article addresses the problem of test-time adaptation of vision-language models. Differently from previous works (e.g., TDA, Karmanov et al. 2024) that use a cache to store samples, the proposed method, DOTA, stores an online estimate of the statistics (mean and variance) of each class of interest.  These statistics are then used during inference to refine standard CLIP predictions.  An active learning strategy exploiting this statistic is also proposed to improve the performance of the model further, asking a user to annotate the least confident examples. Experiments on a wide range of tasks show the efficacy of this approach.

**Strengths:**

1. DOTA revisits principles in the literature on continual learning via nearest class mean classifiers (e.g., [a,b]) for improving the performance of VLMs at test time. Overall the approach is easy to implement and can be considered a valid baseline for future works, performing continuous TTA without the need for storing a cache, as in TDA.

2. The article is well-structured and easy to follow, guiding the reader through all the design choices.

3. DOTA is effective (as shown in the comparisons with TDA, e.g., Tab. 1 and Tab. 2) and computationally cheap (as shown in Tab. 3).

4. Fig. 3 provides an analysis of how the performance of the two models (TDA and DOTA) vary w.r.t. the number of samples, showing the advantages of the latter.

**References**:

[a] Mensink, Thomas, et al. "Distance-based image classification: Generalizing to new classes at near-zero cost." IEEE transactions on pattern analysis and machine intelligence 35.11 (2013): 2624-2637.

[b] Bendale, Abhijit, and Terrance Boult. "Towards open world recognition." Proceedings of the IEEE conference on computer vision and pattern recognition. 2015.

**Weaknesses:**

1. DOTA continually updates its estimates of the statistics. Those might be affected by various factors linked to the experimental protocol, (e.g., order of the classes in the stream, batch size) as well as hyperparameters choice (i.e., the initial variance value, the shrinkage $\epsilon$, $\lambda$'s hyperparameters). Currently, the article does not provide too many insights on these factors, with the analysis mostly limited to the active learning percentile (i.e., Fig. 3). To assess the robustness of the model and provide a thorough study of its performance, it would be interesting to show results across multiple data ordering (i.e., currently it is not clear how many orders have been tested) and whether the performance changes w.r.t. the particular stream considered, even on the edge-cases where the data is non-i.i.d. [c]. Moreover, the hyperparameters may impact the speed of adaptation (e.g., variance initialization, $epsilon$) as well as how much the pretrained model is considered (e.g., $\lambda$s): studying their impact is essential to fully evaluate the complexity of the approach and potential difficulties in applying it on real-world scenarios.

2. While TPT and DiffTPT are strong models for test-time adaptation, they work on the episodic setting, i.e., where adaptation is held out on a single sample, and then the model is reset to its previous state. The possibility of storing/using test-time data (assuming coherence in the sequence) is a non-negligible advantage that DOTA has (and that it shares with TDA). This makes both the "continual adaptation" mark on TPT (Tables 1 and 2) potentially misleading, as well as TDA the only true baseline acting under the same priors of DOTA. To make the results stronger, it would be beneficial to add more baselines, such as DMN [d].

3. Following on the previous points, adapting to an evolving stream is a much more nuanced problem, where correlation between consecutive data may play an important role. Thus, various TTA settings with different types of stream and data dependencies (e.g., practical TTA [e], universal TTA [f]) could have been considered to further show the effectiveness of the approach.

4. A key motivation behind DOTA relies on the test-time forgetting of TDA (lines 52-56). However, there are no experiments demonstrating this point (beyond the quantitative advantages of DOTA). An analysis clearly showing this phenomenon (and how DOTA is more robust to it) would strengthen the motivation behind the approach.

5. The active learning strategy proposed to refine the performance for uncertain samples (Section 3.3) is a nice addition to make the approach more coherent but it lacks competitors. For instance, also TDA could employ a similar strategy (as the update of the cache is based on the confidence of the predictions). Moreover: (i) the accuracy with random feedback is also very close to those achieved with the proposed strategy (e.g., 0.6% gap on average in Tab. 6, 5% percentile); (ii) for the confidence-based scoring to work, the model is assumed to be calibrated, something not always true and that needs a proper discussion [g]; (iii) in Tab. 7 the accuracy of the random baseline is not reported: this is an important reference to put results into perspective.

6. Related work (Section 2) provides a limited discussion on the various types of TTA settings (e.g., [e,f]) as well as on previous methods employing online updates of statistics for continual learning/open world recognition [b] or prototype-based few-shot learning [I,j]. Expanding the discussion would help to better contextualize the work in the current literature.

**Minors**:

- Footnote 1 hints that the model could be applied beyond CLIP. However, there are no experiments confirming this claim. It would have been more thorough to show other models (e.g., SigLIP [h]) to support it.

- Table 4 shows the results only for DOTA. It would be interesting to see the same analysis for the other baselines (e.g., TDA) to contextualize/provide a reference for the results.


**References**:

[a] Mensink, Thomas, et al. "Distance-based image classification: Generalizing to new classes at near-zero cost." IEEE transactions on pattern analysis and machine intelligence 35.11 (2013): 2624-2637.

[b] Bendale, Abhijit, and Terrance Boult. "Towards open world recognition." Proceedings of the IEEE conference on computer vision and pattern recognition. 2015.

[c] Gong, Taesik, et al. "Note: Robust continual test-time adaptation against temporal correlation." Advances in Neural Information Processing Systems 35 (2022): 27253-27266.

[d] Zhang, Yabin, et al. "Dual memory networks: A versatile adaptation approach for vision-language models." Proceedings of the IEEE/CVF conference on computer vision and pattern recognition. 2024.

[e] Yuan, Longhui, Binhui Xie, and Shuang Li. "Robust test-time adaptation in dynamic scenarios." Proceedings of the IEEE/CVF Conference on Computer Vision and Pattern Recognition. 2023.

[f] Marsden, Robert A., Mario Döbler, and Bin Yang. "Universal test-time adaptation through weight ensembling, diversity weighting, and prior correction." Proceedings of the IEEE/CVF Winter Conference on Applications of Computer Vision. 2024.

[g] Tu, Weijie, et al. "An Empirical Study Into What Matters for Calibrating Vision-Language Models." International Conference on Machine Learning. 2024.

[h] Zhai, Xiaohua, et al. "Sigmoid loss for language image pre-training." Proceedings of the IEEE/CVF International Conference on Computer Vision. 2023.

[i] Snell, Jake, Kevin Swersky, and Richard Zemel. "Prototypical networks for few-shot learning." Advances in neural information processing systems 30 (2017).

[j] De Lange, Matthias, and Tinne Tuytelaars. "Continual prototype evolution: Learning online from non-stationary data streams." Proceedings of the IEEE/CVF international conference on computer vision. 2021.

**Questions:**

1. How does the performance change w.r.t. the data stream?
2. What is the impact of the various hyper parameter?
3. How does the test-time forgetting phenomenon happen?
4. How does the model compare with other baselines, e.g., DMN?
5. How does the work relate to existing ones on prototypical networks and the various TTA settings?

---

> ### Author Response · Authors · 2024-11-16
> **Official Response by Authors (1/4)**
>
> We thanks for your valuable suggestions and we will try to address your concerns as follows.
>
>
> ## Weaknesses 1. Factors analysis associated with the experimental protocol.
>
> ### Subweakness a. Hyperparameter analysis.
>
> To better validate the sensitivity of our model to hyperparameters, we conducted systematic experiments and analyses with the following details:
>
>
> 1. On the Selection of Hyperparameter $\sigma$. We tested the values of $\sigma$ in the range \([0.0001, 0.001, 0.002, 0.004, 0.008, 0.02]\) while keeping other parameters fixed. The experimental results show that the model demonstrates strong robustness to different $\sigma$ values. The accuracy (acc) for each value of $\sigma$ is as follows:
>
>      | $\sigma$ | 0.0001 | 0.001 | 0.002 | 0.004 | 0.008 | 0.02  |
>      |------------|--------|-------|-------|-------|-------|-------|
>      | **Acc**    | 70.58  | 70.63 | 70.68 | 70.64 | 70.56 | 70.36 |
>
>    - These results indicate that the impact of $sigma$ on the model's performance is minimal, showcasing its robustness to this parameter.
>
> 2. **On the Selection of Hyperparameters $\eta$ and $\rho$:** Further experiments were conducted by fixing $\sigma$ and testing $\rho$ values in \([0.005, 0.01, 0.02, 0.03]\), while adjusting $\eta$ in \([0.2, 0.3, 0.4, 0.5]\). The results are summarized in the table below:
>
>      | $\eta \backslash \rho$ | 0.005  | 0.01   | 0.02   | 0.03   |
>      |--------------------------|--------|--------|--------|--------|
>      | **0.2**                  | 70.68  | 70.66  | 70.51  | 70.43  |
>      | **0.3**                  | 70.66  | 70.51  | 70.28  | 70.16  |
>      | **0.4**                  | 70.66  | 70.48  | 70.19  | 70.03  |
>      | **0.5**                  | 70.66  | 70.44  | 70.08  | 69.91  |
>
> These results further demonstrate the robustness of our model to the selection of $\eta$ and $\rho$, validating the stability of our proposed method. The above experiments were conducted using the ViT-B/16 backbone on the ImageNet validation set.
>
> ### Subweakness b. Results across multiple data ordering.
>
> We conducted experiments on five test data in different ordering on multiple datasets. The experimental results are shown in the following table. From the experimental results, we can see that the order of test data has little effect on the prediction performance.
>
> |Dataset | Method\Number of Tests   | 1     | 2     | 3     | 4     | 5     | Average |
> |-------------------------- |--------------------------|-------|-------|-------|-------|-------|---------|
> |ImageNet | Dota                        | 70.57 | 70.70 | 70.70 | 70.61 | 70.45 | 70.61   |
> |ImageNet | Dota with 5% human feedback        | 71.09 | 71.10 | 71.00 | 70.85 | 70.90 | 70.99   |
> |ImageNet | Dota with 15% human feedback       | 71.86 | 71.76 | 71.74 | 71.85 | 71.93 | 71.83   |
> |eurosat | Dota                       | 57.78 | 56.99 | 58.15 | 57.28 | 57.22 | 57.48   |
> |eurosat | Dota with 5% human feedback        | 64.89 | 65.38 | 64.74 | 64.60 | 66.28 | 65.18   |
> |eurosat | Dota with 15% human feedback       | 75.68 | 77.90 | 76.85 | 77.57 | 76.93 | 76.99   |
> |OxfordPets| Dota                        | 91.88 | 91.71 | 91.80 | 91.71 | 91.99 | 91.82   |
> |OxfordPets|Dota with 5% human feedback        | 92.56 | 93.05 | 92.78 | 92.89 | 92.56 | 92.77   |
> |OxfordPets | Dota with 15% human feedback       | 93.84 | 93.68 | 93.68 | 93.62 | 93.70 | 93.70   |
>
> ## Weakness 2. More baselines.
>
> We added the following baselines accepted at NeurIPS 2024 most recently. Among the compared methods, BoostAdapter and HisTPT both have the ability of continuous learning. Compared with these methods, the proposed method is still competitive. When DOTA with human feedback is performed, the proposed method is able to achieve even more leading performance.
>
> | Method                | Mean  | Aircraft | Caltech | Cars   | DTD    | EuroSAT | Flowers | Food101 | Pets   | SUN397 | UCF101 |
> |-----------------------|-------|----------|---------|--------|--------|---------|---------|---------|--------|--------|--------|
> | ZERO [1]   | 64.21 | 25.21    | 93.66   | 68.04  | 46.12  | 34.33   | 67.68   | 86.53   | 87.75  | 65.03  | 67.77  |
> | BoostAdapter [2]          | 68.68 | 27.45    | 94.77   | 69.30  | 45.69  | 61.22   | 71.66   | 87.17   | 89.51  | 68.09  | 71.93  |
> | HisTPT [3] | 67.60 | 26.90    | 94.50   | 69.20  | 48.90  | 49.70   | 71.20   | 89.30   | 89.10  | 67.20  | 70.10  |
> | Dota                  | 69.01 | 25.59    | 94.32   | 69.48  | 47.87  | 57.65   | 74.67   | 87.02   | 91.69  | 69.70  | 72.06  |
> | Dota 5% feedback      | 70.96 | 26.73    | 94.56   | 70.95  | 49.82  | 65.00   | 76.86   | 87.17   | 92.78  | 70.49  | 75.26  |
> | Dota 15% feedback     | 73.89 | 28.65    | 95.01   | 73.01  | 53.78  | 76.60   | 79.70   | 87.41   | 93.54  | 71.82  | 79.33  |

---

> ### Author Response · Authors · 2024-11-16
> **Official Response by Authors (2/4)**
>
> ## Weakness 3. Adapting to an evolving stream.
>
> We combined the ImageNet and ImageNet-V2 datasets, first testing the performance on ImageNet and then on ImageNet-V2. This setup evaluates the model's ability to adapt to a continuously changing data stream. The experimental results are shown below.
>
> | Dataset  | Original | Combined |
> |----------|--------------|---------|
> | ImageNet | 70.68        | 70.68   |
> | ImageNet-V2        | 64.41        | 65.09   |
>
> For example, when testing ImageNet-V2 separately, the performance is 64.41, but after integration with ImageNet, the performance slightly improves to 65.09. Investigating test-time adaptation under changing data stream distributions is an interesting direction; however, it may be beyond the scope of this study.
>
> ## Weaknesses 4. Test-time forgetting of TDA.
>
> 1. In principle, when TDA adapts during testing, it will continuously discard the information stored in its cache, which will also cause it to continuously forget information.
> 2. Moreover, we conduct experiments on the ImageNet dataset. When testing on the ImageNet dataset, we record the performance of the most recent 5,000 test samples and compare them with the original zero-shot classifier performance, recording the relationship between the improvement in model performance and the number of test samples seen. The experimental results are shown in the following table. From the experimental results, we can see that **the performance of TDA increases first and then decreases** (due to its forgetting during testing), while the proposed method does not.
>
> | Test Samples Seen (5e^3) | 1    | 2    | 3    | 4    | 5    | 6    | 7    | 8    | 9    | 10   |
> |--------------------------|------|------|------|------|------|------|------|------|------|------|
> | DOTA Improvements        | 0.02 | 1.18 | 1.48 | 1.98 | 1.64 | 2.84 | 2.74 | 2.26 | 2.06 | 2.84 |
> | TDA Improvements         | 0.14 | 0.70 | 0.76 | 1.22 | 0.96 | 0.98 | 1.02 | 0.68 | 0.46 | 0.24 |
>
> ## Weakness 5.  TTA with human feedback
>
> ### Subweakness a. Stronger baseline
> Thanks for your suggestion, we added a stronger baseline for TTA based on human feedback. Please note that since we are the first to define this problem, there are few related methods. Therefore, we compared it with a paper (ATPT[4]) submitted to ICLR at the same time. The experimental results demonstrate that our approach significantly outperforms the proposed method.
>
> | Model                 | Average | Aircraft | Caltech101 | Cars  | DTD   | EuroSAT | Flower102 | Food101 | Pets  | SUN397 | UCF101 |
> |-----------------------|---------|----------|------------|-------|-------|---------|-----------|---------|-------|--------|--------|
> | ATPT with 5% feedback | 67.26   | 24.85    | 94.27      | 67.86 | 48.23 | 49.88   | 72.36     | 86.77   | 90.65 | 67.51  | 70.23  |
> | Ours with 5% feedback | 70.96   | 26.73    | 94.56      | 70.95 | 49.82 | 65.00   | 76.86     | 87.17   | 92.78 | 70.49  | 75.26  |
>
> Moreover, we are also changing the previous comparison method to introduce human feedback and provide a stronger baseline. Specifically, we added an uncertain sample cache on the TDA to maintain the mean of uncertain samples. The experimental results are shown below. The experimental results indicate that the proposed method achieves superior performance. However, TDA may fail in certain cases due to its inefficient utilization of human feedback samples. Specifically, merely storing uncertain samples and their labels in the cache does not lead to performance improvement, possibly because the features of uncertain samples may contain data noise.
>
>
> | Method                | ImageNet | Average except ImageNet | FGVC  | Caltech101 | Cars  | DTD   | EuroSAT | Flower | Food101 | Pets  | SUN397 | UCF101 |
> |-----------------------|----------|---------|-------|------------|-------|-------|---------|--------|---------|-------|--------|--------|
> |TDA 5% Human Feedback     | 69.46    | 65.31   | 23.13 | 91.36      | 64.73 | 41.78 | 55.54   | 69.47  | 85.87   | 89.48 | 64.54  | 67.17  |
> | Ours with 5% feedback | 71.01| 70.96   | 26.73    | 94.56      | 70.95 | 49.82 | 65.00   | 76.86     | 87.17   | 92.78 | 70.49  | 75.26  |
> |TDA 15% Human Feedback    | 69.68    | 67.44   | 23.73 | 91.93      | 66.02 | 44.27 | 64.06   | 70.52  | 85.97   | 90.52 | 65.80  | 71.56  |
> | Ours with 15% feedback |71.83| 73.89|28.65 | 95.01 | 73.01 | 53.78 | 76.60 | 79.70 | 87.41 | 93.54 | 71.82 | 79.33 |

---

> ### Author Response · Authors · 2024-11-16
> **Official Response by Authors (3/4)**
>
> ### Subweakness b. Effectiveness evaluation of selecting uncertain samples based on confidence.
>
> To emphasize the necessity of human feedback, we introduce an additional evaluation metric (ACC*): for samples with human feedback, the labels updated after human feedback are used as the predicted labels to evaluate the accuracy of all test samples. The experimental results are as follows. It can be seen from the experimental results that When confidence-based uncertain sample selection is used, the model performance can be significantly improved.
>
> | Human-Feedback | Method      | Acc   | Acc*  |
> |----------------|-------------|-------|-------|
> | 0%              | Dota       | 70.68 | 70.68 |
> | 5%             | Random      | 70.86 | 72.34 |
> |                | Similarity  | 71.08 | 73.48 |
> |                | Confidence  | 71.01 | 74.52 |
> | 15%            | Random      | 71.28 | 75.61 |
> |                | Similarity  | 71.68 | 78.18 |
> |                | Confidence  | 71.83 | 80.91 |
>
> ### Subweakness c. Regarding confidence calibration.
>
> The paper proposes a confidence-based uncertainty sample selection baseline. It assumes that CLIP is relatively well calibrated. Corresponding analysis has been conducted in some existing works, for example, the Expected Calibration Error of CLIP on the ImageNet dataset is 1.51%[5]. More discussion will be added to the paper.
>
> ### Subweakness d.  The accuracy of the random baseline in Table 7.
>
> We have added the experimental results. The reason we do not report a random sampling baseline is that when the amount of test sample data is large enough, the accuracy of random sampling should be close to the accuracy of the zero-shot classifier.
>
> | Feedback Percentile | Method       | FGVC   | Caltech101 | Cars   | DTD    | EuroSAT | Flower  | Food101 | Pets   | SUN397 | UCF101 | Average |
> |---------------------|--------------|--------|------------|--------|--------|---------|---------|---------|--------|--------|---------|---------|
> | /                   | Baseline     | 24.66  | 94.30      | 65.93  | 44.59  | 48.22   | 70.12   | 85.91   | 88.59  | 66.98  | 65.75  | 64.59   |
> | 5%                  | Random       | 19.17  | 84.09      | 51.14  | 47.54  | 37.06   | 71.59   | 76.20   | 93.33  | 67.72  | 62.62  | 61.05   |
> |                     | Similarity   | 19.32  | 91.95      | 51.76  | 30.36  | 5.00    | 42.86   | 54.56   | 50.00  | 55.61  | 32.22  | 43.36   |
> |                     | Confidence   | 11.80  | 68.35      | 25.00  | 15.87  | 20.51   | 9.63    | 31.74   | 37.93  | 19.79  | 18.09  | 25.87   |
> | 15%                 | Random       | 17.16  | 84.81      | 58.90  | 44.72  | 38.53   | 65.67   | 77.50   | 89.72  | 63.71  | 58.94  | 59.97   |
> |                     | Similarity   | 21.37  | 95.74      | 58.94  | 32.70  | 13.73   | 37.89   | 63.40   | 65.74  | 55.91  | 45.68  | 49.11   |
> |                     | Confidence   | 11.36  | 71.81      | 29.81  | 18.12  | 19.63   | 20.16   | 44.91   | 52.04  | 30.66  | 21.42  | 39.11   |
>
> ## Minor weakness 1. Experiments on other models.
>
>
> #### 1. Experimental Setup:
> - **Backbone:** PLIP[1]
> - **Baseline:** We used PLIP's predictions as the baseline.
>
> #### 2. Datasets:
> We evaluated our method on the following medical image datasets, where labels were converted into descriptive sentences (e.g., converting "tumor" into "H&E image of a tumor") for evaluation. The model was tested on the test dataset without additional fine-tuning.
> 1. **Kather colon dataset** (9 different tissue types).
> 2. **PanNuke** dataset (benign vs. malignant).
> 3. **WSSS4LUAD** dataset (tumor vs. normal).
>
> #### 3. Unconfident Sample Selection Method:
> We adopted **confidence-based selection** for identifying low-confidence samples for human feedback.
>
> #### 4. Experimental Results:
> The following table summarizes the accuracy results. Note that "Human Feedback" indicates the accuracy achieved with the default evaluation setting, while "Human Feedback*" reflects the accuracy when all samples with ground truth labels are considered correct, as per the reviewer's suggestion.
>
> | Dataset          | Kather  | PanNuke | WSSS4LUAD | Average |
> |------------------|---------|---------|-----------|---------|
> | **PLIP (Baseline)**      | 45.60   | 71.56   | 70.31     | 62.49   |
> | **DOTA**                 | 55.22   | 72.25   | 72.32     | 66.60   |
> | **5% Human Feedback**    | 56.52   | 72.35   | 72.62     | 67.16   |
> | **5% Human Feedback***   | 57.82   | 73.83   | 73.25     | 68.30   |
> | **15% Human Feedback**   | 58.32   | 72.46   | 72.79     | 67.86   |
> | **15% Human Feedback***  | 61.60   | 76.91   | 74.41     | 70.97   |
>
> ---

---

> ### Author Response · Authors · 2024-11-16
> **Official Response by Authors (4/4)**
>
> ## Minor weakness 2. We added the experimental results of TDA in Table 4.
>
> We ran the TDA code in our local environment and reported the performance on the entire test data and the last 50% of the samples. The experimental results are shown below. From the experimental results, we can see that the performance of the last 50% of samples of our method is almost higher than that of all test samples, which shows that the performance of our method is improving. However, TDA is different, and its performance has declined on the caltech101, cars, and pets datasets.
>
> | dataset                         | fgvc  | caltech101 | cars  | dtd   | flower | food101 | pets  | sun397 | ucf101 |
> |---------------------------------|-------|------------|-------|-------|--------|---------|-------|--------|--------|
> | TDA (All test samples)          | 25.59 | 94.16      | 67.35 | 45.98 | 71.66  | 86.02   | 90.00 | 67.62  | 71.16  |
> | TDA (The last 50% of test samples) | 26.57 | 93.59      | 66.95 | 46.22 | 71.75  | 86.02   | 89.26 | 67.86  | 72.20  |
> | Dota (All test samples)         | 25.59 | 94.32      | 69.48 | 47.87 | 74.67  | 87.02   | 91.69 | 69.70  | 72.06  |
> | Dota (The last 50% of test samples) | 27.11 | 94.65      | 69.88 | 50.95 | 75.89  | 87.10   | 93.02 | 70.67  | 73.20  |
>
> [1] Frustratingly Easy Test-Time Adaptation of Vision-Language Models
>
> [2] BoostAdapter: Improving Vision-Language Test-Time Adaptation via Regional Bootstrapping
>
> [3] Historical Test-time Prompt Tuning for Vision Foundation Models
>
> [4] Active test time prompt learning in vision-language models
>
> [5] Open-Vocabulary Calibration for Fine-tuned CLIP
>
> [6] Pathology Language and Image Pre-Training

---

> > ### Comment · Reviewer_DNWr · 2024-11-18
> > **Thanks, further clarifications**
> >
> > I thank the authors for the thorough responses, answering most of the concerns raised in the initial review(s). I would like to ask for further clarifications (reference numbering is kept as above):
> >
> > - W1: Thanks for analyzing the hyperparameters! Regarding the ordering of the stream in the data, does the performance change in cases where the order is non-i.i.d. (see [c] above)? This would show the robustness of the approach beyond the random orderings of the streams.
> >
> > - W2: Thanks for adding very recent baselines. Is there a reason for not including DMN [d]?  Moreover, in the natural distribution case (Tab. 1) is there a reason for not reporting results on ImageNet-v2 (as done in other works, e.g., TDA)?
> >
> > - W5: Thanks for the clarifications. One point I wanted to make (and I apologize for my lack of clarity) is that, when proposing an active learning approach for TTA, one should show that existing active learning strategies do not work in this setting/are less effective. This is currently not shown in the paper, where the tables (Tab. 6 and Tab. 7) mostly focus on ablations of different scoring functions (e.g., similarity vs confidence). It would be interesting to discuss/test the limitations of other existing approaches and their simple adaptation (e.g., entropy-based [k],  or even those based on diversity measures [l,m]). Another point for the discussions can also be [n] for VLMs.
> >
> > - Minor weakness 1: I thank the authors for this effort, including another backbone and other datasets. While the reported results are interesting, TDA should be applied to the same model to provide a reference baseline/competitor.
> >
> > **References:**
> >
> > [k] Holub, Alex, Pietro Perona, and Michael C. Burl. "Entropy-based active learning for object recognition." CVPR 2008.
> >
> > [l] Sener, Ozan, and Silvio Savarese. "Active Learning for Convolutional Neural Networks: A Core-Set Approach." ICLR 2018.
> >
> > [m] Ash, Jordan T., et al. "Deep Batch Active Learning by Diverse, Uncertain Gradient Lower Bounds." ICLR 2024.
> >
> > [n] Bang, Jihwan, Sumyeong Ahn, and Jae-Gil Lee. "Active Prompt Learning in Vision Language Models." CVPR 2024.

---

> > > ### Author Response · Authors · 2024-11-19
> > > **Further clarifications by Authors (1/3)**
> > >
> > > ## W1 Performance change in cases where the order is non-i.i.d.
> > >
> > > Firstly, we add the experiments you proposed regarding the changes in data distribution during testing. Specifically, consistent with the baseline settings in paper [c], we conducted experiments on the ImageNet dataset. We used the Dirichlet distribution to simulate non-independent and identically distributed (non-i.i.d.) data streams on the ImageNet dataset. Specifically, we divided the ImageNet dataset into multiple time slices and adjusted the concentration parameter of the Dirichlet distribution to generate varying degrees of non-i.i.d. data distributions. **Across different time slices and $\alpha$ settings, it can be observed that a small $\alpha$ leads to a slight decrease in accuracy. However, the decrease is minimal, indicating that the DOTA model exhibits strong robustness to Non-I.I.D. data streams.**
> > >
> > >
> > > **Time Slices:** We selected time slice quantities of {5, 10}, representing data being incrementally input to the model in distinct "time batches." Each time slice contains a varying number of samples and class distributions.
> > >
> > > **Concentration Parameter ($[\alpha]_K$):** The concentration parameter of the Dirichlet distribution controls the uniformity of $K$-class distribution within each time slice. Specifically, at each time slice, we randomly sample a class proportion from the Dirichlet distribution, which determines the data distribution for that particular time slice. Smaller $\alpha$ values (e.g., 0.1) produce highly uneven distributions, while larger $\alpha$ values (e.g., 0.5 and 1) result in more uniform distributions.
> > >
> > > **Evaluation Setting**. Since the size of each time-slice sub-dataset is unequal, the final average accuracy is not the mean of the accuracies from each time slice but the weighted average based on the number of samples, effectively representing the overall mean accuracy across all samples.
> > >
> > > 1. **Experimental Results**.
> > >
> > > | Time Slices | Alpha | 1     | 2     | 3     | 4     | 5     | Average |
> > > |-------------|-------|-------|-------|-------|-------|-------|---------|
> > > | 5           | 0.1   | 68.67 | 69.72 | 71.60 | 71.14 | 71.08 | 70.45   |
> > > | 5           | 0.5   | 69.57 | 70.75 | 69.53 | 71.85 | 70.83 | 70.52   |
> > > | 5           | 1     | 69.47 | 71.59 | 71.06 | 69.92 | 71.83 | 70.83   |
> > >
> > > | Time Slices | Alpha | 1     | 2     | 3     | 4     | 5     | 6     | 7     | 8     | 9     | 10    | Average |
> > > |-------------|-------|-------|-------|-------|-------|-------|-------|-------|-------|-------|--------|--------|
> > > | 10          | 0.1   | 70.44 | 69.48 | 67.26 | 70.25 | 71.49 | 68.91 | 73.03 | 70.48 | 69.72 | 72.39 | 70.39  |
> > > | 10          | 0.5   | 69.01 | 69.30 | 72.10 | 71.04 | 69.83 | 69.90 | 70.30 | 70.63 | 72.34 | 70.81 | 70.55  |
> > > | 10          | 1     | 67.23 | 70.12 | 71.01 | 68.25 | 69.76 | 71.14 | 72.01 | 72.59 | 71.46 | 71.75 | 70.66  |
> > >
> > >
> > > 2. **Theoretical discussion in cases of relatively mild changes in the test data distribution** (which I believe may be more common in practice), our approach is theoretically well-suited to address such scenarios for two reasons. First, the method can incorporate human feedback, enabling timely intervention to provide accurate labels when the distribution changes. Second, the method allows for online estimation of the data distribution, enabling the model to adapt to distribution shifts (theoretically). Additionally, some simple modifications could be proposed to further improve our method in handling distribution shift issues. For instance, incorporating a sliding window mechanism to mitigate the impact of samples that are significantly distant from the current test data during the distribution estimation process.
> > >
> > > 3. **Theoretical discussion in cases of more extreme shifts**. We recognize that in cases of more extreme shifts in data distribution, the performance of the model may be affected. Theoretically, accurate distribution estimation during testing requires a sufficient number of samples from the same distribution to achieve a minimal regret error bound. In other words, only when there are sufficient samples from the same distribution can we reliably estimate the data distribution. We will add this issue to the limitations section of the paper.
> > >
> > > 4. Finally, we would like to respectfully acknowledge that, similar to other comparative methods such as TDA, our approach is currently designed with a focus on specific deployment scenarios where the data distribution remains relatively stable. Adapting to scenarios with significant test distribution shifts is indeed a challenging task, and we appreciate the reviewer’s insight on this problem. While the proposed method may not yet fully address such cases, we believe this does not detract from its strong performance in scenarios with more stable deployment conditions. Thank you for your thoughtful feedback, which has been invaluable in helping us further reflect on and refine our work.

---

> > > ### Author Response · Authors · 2024-11-19
> > > **Further clarifications by Authors (2/3)**
> > >
> > > ## W2 Questions about new baselines.
> > >
> > > ### Subweakness a. Reason for not including DMN as baseline.
> > >
> > > Thank you for your suggestion. The primary reason we did not include DMN as a comparative method is that it is not fully aligned with our original experimental setup. Specifically, **DMN incorporates prompts generated by large language models, whereas our approach uses the standard prompts proposed in the CLIP paper.** As noted in the DMN paper: “We follow existing works to conduct the image split in few-shot learning and adopt the textual prompt in [43, 68],” where reference [43] refers to "What does a platypus look like? Generating customized prompts for zero-shot image classification". This makes a direct comparison between our method and DMN somewhat unfair. Therefore, we introduced three other more recent baselines with same experimetal setting as comparative methods instead.
> > >
> > >
> > > ### Subweakness b. Reason for not reporting results on ImageNetV2.
> > >
> > > We presented the performance on ImageNetV2 in the limitations section of the original paper's appendix. We excluded it from the evaluation because the number of test samples per class in ImageNetV2 is too small to be suitable for the method proposed in this paper.  Specifically, While our approach demonstrates the advantage of continuously estimating the distribution of test data, enabling adaptation to the test data, it does not consistently outperform TDA across all datasets. For example, on the ImageNetV2 dataset with only 10 samples per class, Dota does not significantly exceed TDA. The potential reason for this is that the proposed method still requires a certain number of samples to enhance the reliability of distribution estimation during testing. However, its performance on the last 50% of the test samples shows a clear improvement. This suggests that the proposed model has the potential to further improve as more test samples become available. The specific performance results are shown as follows.
> > >
> > >
> > > | Method                     | ViT-B/16 | ResNet-50 |
> > > |----------------------------|----------|-----------|
> > > | CLIP                       | 61.88    | 52.91     |
> > > | TDA                        | 64.67    | 55.54     |
> > > | *Dota* (All test samples)  | 64.41    | 55.27     |
> > > | *Dota* (The last 50% of test samples) | 65.06 | 55.82 |

---

> > > ### Author Response · Authors · 2024-11-19
> > > **Further clarifications by Authors (3/3)**
> > >
> > > ## Weakness 5. the limitations of other existing approaches and their simple adaptation
> > >
> > > | **Feature**          | **Test-Time Adaptation**                           | **Active Learning**                           |
> > > |-----------------------|---------------------------------------------------|-----------------------------------------------|
> > > | **Data Access**       | Sequential stream, samples seen one at a time.    | Large pool of unlabeled data available.       |
> > > | **Selection Method**  | Real-time decision, no revisiting samples.        | Global scoring of all samples, iterative.     |
> > >
> > > **Key differences in sample selection strategies between active learning and the proposed method (Main contribution of the proposed paradigm)**. In the contexts of test-time adaptation and active learning, the core difference in collecting human feedback lies in the data acquisition method. Test-time adaptation involves selecting high-value samples from a continuous data stream. In contrast, traditional active learning typically assumes access to a small labeled dataset alongside a large pool of unlabeled data. In the active learning framework, sample selection involves scoring all unlabeled samples and choosing the most valuable subset for labeling. In the test-time adaptation setting, however, data is presented sequentially as an unlabeled stream, where each sample is available only during testing and cannot be revisited afterward. This constraint necessitates quick decisions during testing about whether a given sample warrants labeling, without prior access to the entire pool of unlabeled samples. Consequently, sample selection strategies in test-time adaptation must rely not only on the score of the current sample but also on information gathered from previously observed samples.
> > >
> > > **Similarities between active learning and test-time adaptation with human-feedback**. Both approaches involve evaluating or scoring samples to determine their value. The criteria for scoring might include uncertainty[7,K], diversity[I,M], confidence[9] or representativeness[8] of the sample. In the current version, we adopted the simplest and most effective approach: scoring the current sample based on confidence to determine whether to collect its label. However, this confidence-based scoring method is not the primary contribution of this work. We acknowledge that other criteria may also be applicable. For instance, we have also experimented with sample performance using similarity-based and entropy-based scoring methods.
> > >
> > > **Active Label Acquisition for Vision-Language Models**. Thank you for your suggestion. In paper [n], we observed a phenomenon where using active learning methods to select samples for vision-language models results in class imbalance among the selected samples. We conducted experiments in the context of test-time adaptation and found similar patterns. The experimental results are as follows:
> > >
> > > - **[0, 300]:** 300 classes receive no feedback.
> > > - **[1, 203]:** 203 classes receive exactly one ground truth label.
> > > - **[2–6, 390]:** 390 classes receive between 2 and 6 ground truth labels.
> > > - **[7–11, 92]:** 92 classes receive between 7 and 11 ground truth labels.
> > > - **[12–16, 15]:** 15 classes receive between 12 and 16 ground truth labels.
> > >
> > > These findings highlight the potential for further refinement of the methods. Addressing the observed class imbalance during active sample selection and feedback acquisition could further enhance the effectiveness of our approach. We leave this as future work, as it may be beyond the main contribution of this paper. More disscussions will be added in the final version of th paper.
> > >
> > > ## Minor weakness 1.
> > > Thank you for your reply, we have added the corresponding experimental results. The experimental results demonstrate that the proposed method significantly outperforms TDA.
> > >
> > > | Method            | Kather | PanNuke | WSSS4LUAD | Average |
> > > |--------------------|--------|---------|-----------|---------|
> > > | plip(baseline)    | 45.60  | 71.56   | 70.31     | 62.49   |
> > > | TDA               | 49.39  | 71.56   | 72.13     | 64.36   |
> > > | dota              | 55.22  | 72.25   | 72.32     | 66.60   |
> > >
> > >
> > > [7] How to measure uncertainty in uncertainty sampling for active learning.
> > >
> > > [8] Exploring representativeness and informativeness for active learning
> > >
> > > [9] Confidence-based active learning

---

> > > > ### Comment · Reviewer_DNWr · 2024-11-22
> > > >
> > > > I thank the authors for their thorough answers, addressing the remaining points. I am willing to increase my score, but I will do it once the revised manuscript is uploaded, confirming the promised changes have been implemented.
> > > >
> > > > I hope these points of discussion will be included. I also suggest including the results on ImageNet-V2  in the main tables. Obtaining negative results and failure cases is part of the research process: discussing them can only add value to the manuscript and hint at potential future directions.

---

> > > > > ### Author Response · Authors · 2024-11-24
> > > > >
> > > > > We  greatly appreciate your prompt response and invaluable feedback. Following your suggestions, we have made revisions to the manuscript. Please kindly note that, due to time constraints and the page limitations of this draft, the current version remains a draft, and we will diligently continue refining it to ensure it meets the highest standards.
> > > > >
> > > > > We also sincerely wish you great success with your own research.

---

> > > > > ### Author Response · Authors · 2024-11-27
> > > > > **Any concerns about our manuscript**
> > > > >
> > > > > Dear Reviewer, since November 27th is the final deadline for submitting the manuscript, I would like to know if you have any concerns about our manuscript that we could address through revisions to improve its score.

---

> > > > > > ### Comment · Reviewer_DNWr · 2024-11-27
> > > > > >
> > > > > > Dear authors,
> > > > > >
> > > > > > Thanks for the message. I am satisfied with the current version and increased my score. After discussing it with the other reviewers, I will update the justification (and eventually the score).
> > > > > >
> > > > > > As a suggestion, it would be good to make the comparisons consistent between Tab. 1 and Tab. 3, if possible (now there is a mismatch among the reported baselines). Also, please double-check the manuscripts for typos, etc. (e.g., row 440).

---

> > > > > > > ### Author Response · Authors · 2024-11-27
> > > > > > > **Thank for your efforts**
> > > > > > >
> > > > > > > Dear reviewer, we deeply appreciate your review, feedback, and discussion with other reviewers. We promise to address the shortcomings in our manuscript, which may have arisen due to time constraints or other factors, and will make every effort to improve it to the highest standard.

---

### Official Review · Reviewer_xgaE · 2024-11-02

**Soundness:** 3
**Presentation:** 3
**Contribution:** 3
**Rating:** 6
**Confidence:** 5

**Summary:**

The author presents DOTA, a method that adapts to deployment conditions by continually estimating test sample distributions rather than memorizing them. Using Bayes’ theorem, Dota computes posterior probabilities for real-time adaptation. A human-in-the-loop feature also gathers feedback on uncertain samples, enhancing adaptability. Experiments show Dota outperforms current methods with continuous test-time learning.

**Strengths:**

1. This paper is well written and easy to follow.
2. The idea is interesting,  the motivation of this paper is clear as well as the novelty of the method.
3. The proposed method is extensively tested against prior work and outperforms on a variety of tasks/baselines.

**Weaknesses:**

1. This paper does not include a detailed case study focused on a particular domain or a challenging dataset.
2. This paper does not include experiments assessing the model's sensitivity to hyperparameters. A detailed analysis of hyperparameter tuning could offer valuable insights into the robustness and generalizability of the proposed approach.
3. The paper could benefit from additional visualizations illustrating Test-time adaption with human feedback

**Questions:**

1. This paper is novel and interesting; would you consider making the code open source?
2. In Table 1, for ResNet-50, DOTA’s performance is only slightly better than TDA, with an average improvement of just 0.15%. I would like to understand the reason for this marginal gain.

---

> ### Author Response · Authors · 2024-11-16
> **Official Response by Authors (1/2)**
>
> ## Weakness 1. Detailed case study focused on a particular domain or a challenging dataset.
> We did not use other more challenging datasets initially to maintain a comparison with previous TTA works. Based on your suggestion, we selected three challenging dataset in the medical domain for evaluation. Below, we provide the details of the experimental setup, datasets, and results:
>
> #### 1. Experimental Setup:
> - **Backbone:** PLIP[1]
> - **Baseline:** We used PLIP's predictions as the baseline.
>
> #### 2. Datasets:
> We evaluated our method on the following medical image datasets, where labels were converted into descriptive sentences (e.g., converting "tumor" into "H&E image of a tumor") for evaluation. The model was tested on the test dataset without additional fine-tuning.
> 1. **Kather colon dataset** (9 different tissue types).
> 2. **PanNuke** dataset (benign vs. malignant).
> 3. **WSSS4LUAD** dataset (tumor vs. normal).
>
> #### 3. Unconfident Sample Selection Method:
> We adopted **confidence-based selection** for identifying low-confidence samples for human feedback.
>
> #### 4. Experimental Results:
> The following table summarizes the accuracy results. Note that "Human Feedback" indicates the accuracy achieved with the default evaluation setting, while "Human Feedback*" reflects the accuracy when all samples with ground truth labels are considered correct, as per the reviewer's suggestion.
>
>
> | Dataset          | Kather  | PanNuke | WSSS4LUAD | Average |
> |------------------|---------|---------|-----------|---------|
> | **PLIP (Baseline)**      | 45.60   | 71.56   | 70.31     | 62.49   |
> | **TDA**               | 49.39  | 71.56   | 72.13     | 64.36   |
> | **DOTA**                 | 55.22   | 72.25   | 72.32     | 66.60   |
> | **5% Human Feedback**    | 56.52   | 72.35   | 72.62     | 67.16   |
> | **5% Human Feedback***   | 57.82   | 73.83   | 73.25     | 68.30   |
> | **15% Human Feedback**   | 58.32   | 72.46   | 72.79     | 67.86   |
> | **15% Human Feedback***  | 61.60   | 76.91   | 74.41     | 70.97   |
>
> ---
>
> #### 5. Observations:
> The proposed method (DOTA) with human feedback consistently improves accuracy compared to the baseline (PLIP) across datasets, especially when using the suggested evaluation setting (Human Feedback*).
>
> ## Weakness 2. Hyperparameter analysis.
>
> To better validate the sensitivity of our model to hyperparameters, we conducted systematic experiments and analyses with the following details:
>
>
> 1. On the Selection of Hyperparameter $\sigma$. We tested the values of $\sigma$ in the range \([0.0001, 0.001, 0.002, 0.004, 0.008, 0.02]\) while keeping other parameters fixed. The experimental results show that the model demonstrates strong robustness to different $\sigma$ values. The accuracy (acc) for each value of $\sigma$ is as follows:
>
>      | $\sigma$ | 0.0001 | 0.001 | 0.002 | 0.004 | 0.008 | 0.02  |
>      |------------|--------|-------|-------|-------|-------|-------|
>      | **Acc**    | 70.58  | 70.63 | 70.68 | 70.64 | 70.56 | 70.36 |
>
>    - These results indicate that the impact of $sigma$ on the model's performance is minimal, showcasing its robustness to this parameter.
>
> 2. **On the Selection of Hyperparameters $\eta$ and $\rho$:** Further experiments were conducted by fixing $\sigma$ and testing $\rho$ values in \([0.005, 0.01, 0.02, 0.03]\), while adjusting $\eta$ in \([0.2, 0.3, 0.4, 0.5]\). The results are summarized in the table below:
>
>      | $\eta \backslash \rho$ | 0.005  | 0.01   | 0.02   | 0.03   |
>      |--------------------------|--------|--------|--------|--------|
>      | **0.2**                  | 70.68  | 70.66  | 70.51  | 70.43  |
>      | **0.3**                  | 70.66  | 70.51  | 70.28  | 70.16  |
>      | **0.4**                  | 70.66  | 70.48  | 70.19  | 70.03  |
>      | **0.5**                  | 70.66  | 70.44  | 70.08  | 69.91  |
>
> These results further demonstrate the robustness of our model to the selection of $\eta$ and $\rho$, validating the stability of our proposed method. The above experiments were conducted using the ViT-B/16 backbone on the ImageNet validation set.

---

> ### Author Response · Authors · 2024-11-16
> **Official Response by Authors (2/2)**
>
> ## Weakness 3. visualizations and analyzation of uncertain samples.
>
> Thank you for your suggestion to provide additional visualizations illustrating test-time adaptation with human feedback. To address this, we conducted an analysis on ImageNet to investigate the distribution of human feedback across different classes and to determine whether the feedback is concentrated in specific categories. We included samples with 5% and 15% human feedback for this analysis.
>
> From the results, it can be observed that when the human feedback rate is set to 5%, approximately half of the categories receive at least one ground truth label from human feedback. However, the performance improvement due to human feedback remains relatively consistent even when only a limited number of samples per category are labeled.
>
> Below is the distribution of human feedback under a 5% feedback setting:
>
> [0, 300]: 300 classes receive no feedback.
> [1, 203]: 203 classes receive exactly one ground truth label.
> ['2–6', 390]: 390 classes receive between 2 and 6 ground truth labels.
> ['7–11', 92]: 92 classes receive between 7 and 11 ground truth labels.
> ['12–16', 15]: 15 classes receive between 12 and 16 ground truth labels.
>
> This analysis highlights two key observations:
>
> - The proposed method achieves stable performance improvements even with an uneven distribution of human feedback across categories.
> - The more challenging samples tend to correlate with class imbalance.
>
> ## Question 1. Availability of code.
> We will definitely open source our code, and in fact our code is already ready.
>
> ## Question 2. Why is DOTA's improvement on ResNet small?
>
> The potential reason lies in the difference in image representation dimensions obtained by ResNet and ViT-B/16. Specifically, the representation dimension of ResNet is 1024, whereas for ViT-B/16, it is 512. In our method, this disparity results in a significant change in the number of parameters required for estimating the test data distribution. Without any simplification, the number of distribution parameters that need to be estimated for each category is $D(D+1)$, where $D$ is the representation dimension. Consequently, the number of parameters for ResNet is approximately four times that of ViT-B/16. This difference may explain the slight improvement in model performance when using the ResNet backbone.
>
> [1] Pathology Language and Image Pre-Training

---

> > ### Comment · Reviewer_xgaE · 2024-11-26
> >
> > I would like to thank the authors for their responses to my concerns. They have already conducted the required experiments, and the results demonstrate the effectiveness of the proposed method. Additionally, the authors are preparing to release the code for this paper. Therefore, I would like to maintain my original score

---

> > > ### Author Response · Authors · 2024-11-27
> > > **The code is available now.**
> > >
> > > Dear reviewer, I appreciate your positive response. In order to make the code publicly available and assist in understanding our methodology, we have organized and uploaded our code as supplementary material on the OpenReview platform.

---

### Official Review · Reviewer_QFMN · 2024-11-02

**Soundness:** 2
**Presentation:** 3
**Contribution:** 3
**Rating:** 6
**Confidence:** 5

**Summary:**

The authors address the problem of Test Time Adaptation of Vision Langugage models. They propose to continuously estimate the distribution of test samples, which they leverage through Bayes theorem, to make the final test predictions. They also collect human feedback to receive supervision for uncertain samples.

**Strengths:**

- They propose to estimate the class distributions in an online manner.
- The adaptive fusion of zero shot text classifier based predictions and the distribution based feature similarities is simple and intuitive.
- This being a backpropogation free approach, is very light-weight computationally, which is a great advantage for TTA.
- The paper is well written and easy to follow, however several clarifications are required.

**Weaknesses:**

1. **Distributional Test Time Adaptation:** In a single image TTA setting, the whole section 3.2 is quite unclear. There is no **batch** of samples in this setting. So, how are the class distributions actually updated at each time step.

2. **Samples for distribution estimation:** Are all test samples used for updating the class distributions? Wouldn't the use of low confident/uncertain samples lead to bad parameter estimates in eqn(4)?

3. **TTA with human feedback:** While this is one of the major contribution of the paper, it appears to only result in modest improvements. 5% and 15% is a lot of data to ask labels for, from a human. However, the results improve only of the order of 1-2%. This makes the efficiency of this whole process questionable.

4. **Performance evaluation of TTA with human feedback:** As the test samples arrive in an online manner and based on uncertainty, how is the final accuracy evaluated here? Are these samples inclusive when evaluating accuracy? If so, you should be using the ground truth as predictions for actively labeled samples. Then the accuracy should be up by about 5% or 15%. As this is not the case in the results reported, are the labeled samples excluded from evaluation? This needs to be clarified. For fair comparison, all results should be reported on the complete test set, even when using human feedback.

5. **Need stronger baselines for TTA with human feedback:** To study the role of TTA with human feedback, stronger baselines need to be established, using different selection strategies, like random, confidence, entropy etc. and report the accuracy of complete test set. As all strategies have same amount of labeled samples included, the performance improvement due these strategies as well as the gains wrt no human feedback can be assessed.

6. **Amount of human feedback:** 5% and 15% is a lot of supervision and this may not be feasible during test time. It's more practical to ask labels for about 1-2% of test data. Experiments with stronger baselines, with lesser supervision, with correct evaluation method, is required to actually understand and evaluate the role of human feedback in TTA.

7. **Choice of hyperparameters:**
In Implementation details, it is mentioned that validation sets are used to choose the hyperparameters. However, in TTA, one does not have access to any data from the test distribution apriori. Hence, validation data is not accessible in practice. Well, if one had access to validation data for test data, it provides a lot more information and could be used for more than just hyperparameter tuning.

**Questions:**

1. **Test Distribution Estimation:** In line 212 and from equations (4) and (6), it is described such that a batch of test samples arrive at each time step. However, prior baselines TPT, TDA perform single image TTA. Further, in Implementation details, in line 706, it is mentioned batch size is 1. Please clarify if single image TTA is done here as well, for fair comparison. If so, what does $n$ refer to in equations (5) and (6). How are $\mu_k$ and $\sigma_k$ estimated in single image TTA. Are you storing features, as done in TTA as well, along with the statistics?

2. **Selection of uncertain samples:**
Confidence is softmax applied over the similarity scores only right? Why is there such a large discrepancy using these two similar metrics (Table 7)? Also, is this similarity and confidence estimated from zero shot classifier or the classifier proposed? And why'd you choose what you choose?

3. **Sensitivity to hyperparameters:** How sensitive is the method on the choice of the parameters $\sigma, \eta, \rho$, as it's not practical to assume access to validation data before actually doing TTA?

4. **Human in the loop TTA:** Please refer to the weaknesses and address the relevant concerns raised.

---

> ### Author Response · Authors · 2024-11-16
> **Official Response by Authors (1/3)**
>
> We thanks for your valuable suggestions and we will try to address your concerns as follows. We are eager to engage in a more detailed discussion with you.
>
> ## **Weakness 1**. How to conduct Distributional Test Time Adaptation (DOTA).
>
> As shown in Fig. 1 in the paper, DOTA dynamically updates distributions of different classes during testing. Specifically, DOTA always maintains the distribution information of different classes (i.e., mean and covariance matrix) during testing, and updates its distribution information based on its representation information after obtaining new samples (Eq. 6 in the manuscript).
>
> **The setting of DOTA**. Benefiting from the test-time adaptation (TTA) framework, which leverages data distribution estimation without altering the original model parameters or prompts, and utilizing a vectorized distribution estimation strategy, the proposed method is inherently independent of test batch size. This flexibility arises because the vectorized approach allows the model to effectively adapt to the data distribution across multiple instances simultaneously, enabling it to work seamlessly in both single-image TTA settings (batch size = 1) and multi-image TTA settings (batch size > 1). In practice, to be consistent with the comparison method, we set the batch size to 1.
>
> ## **Weakness 2**. Samples for distribution estimation.
> 1. When there is no human feedback, all test samples are used to update the data distribution.
> 2. In practice, however, the potential parameter estimation errors caused by this data update method are controllable, thanks to the implicit weighting process in the update process (higher confidence samples have greater weights, and vice versa).
> 3. In principle, the current data distribution update process can be seen as a traditional EM algorithm. The EM algorithm follows the principle of maximum likelihood estimation, i.e., estimating model parameters by selecting parameter values that make the observed data most likely to occur.
> 4. To further avoid the negative impact of low confidence samples, we propose human-feedback-based TTA. This approach updates the test data distribution parameters by detecting low confidence samples and collecting human feedback.
>
> ## **Weaknesses 3 & 4**. How to conduct TTA with human feedback.
>
> 1. Evaluation. To ensure that the proposed method is comparable with other methods in performance (using the same number of test samples), we evaluate accuracy during testing by using the predicted labels from before collecting human feedback, even for samples where human feedback is collected.
> 2. More evaluation metrics. To emphasize the necessity of human feedback, we introduce an additional evaluation metric (ACC*): for samples with human feedback, the labels updated after human feedback are used as the predicted labels to evaluate the accuracy of all test samples. The experimental results are as follows. It can be seen from the experimental results that when human feedback is used, the model performance can be significantly improved. Also note that since we are not always able to accurately identify misclassified samples (another difficult task), the accuracy improvement may be smaller than the proportion of human feedback.
>
> | Human-Feedback | Acc   | Acc*  |
> |----------------|-------|-------|
> | 0%             | 70.68 | 70.68 |
> | 5%             | 71.01 | 74.52 |
> | 15%            | 71.83 | 80.91 |
> 3. Our method is a baseline version of TTA based on human feedback (as far as we know, we are the first to propose this paper that is very meaningful in real-world deployment), how to better collect human feedback, and based on human feedback further improving performance is a goal that needs attention in the future.

---

> ### Author Response · Authors · 2024-11-16
> **Official Response by Authors (2/3)**
>
> ## Weakness 5. Stronger baselines for TTA
>
> We enhance the baseline from following three perspectives.
> 1. We incorporate more sample selection strategies. The experimental results are shown in the table below. From the experimental results, it can be seen that the confidence-based selection can achieve better performance.
>
> | Human-Feedback | Method      | Acc   | Acc*  |
> |----------------|-------------|-------|-------|
> | 0%              | Dota       | 70.68 | 70.68 |
> | 5%             | Random      | 70.86 | 72.34 |
> |                | Similarity  | 71.08 | 73.48 |
> |                | Confidence  | 71.01 | 74.52 |
> | 15%            | Random      | 71.28 | 75.61 |
> |                | Similarity  | 71.68 | 78.18 |
> |                | Confidence  | 71.83 | 80.91 |
>
> 2.We include another comparison method (ATPT)[1], which, like ours, is currently under review for ICLR 2025. The experimental results demonstrate that our approach significantly outperforms the proposed method.
>
> | Model                 | Average | Aircraft | Caltech101 | Cars  | DTD   | EuroSAT | Flower102 | Food101 | Pets  | SUN397 | UCF101 |
> |-----------------------|---------|----------|------------|-------|-------|---------|-----------|---------|-------|--------|--------|
> | ATPT with 5% feedback | 67.26   | 24.85    | 94.27      | 67.86 | 48.23 | 49.88   | 72.36     | 86.77   | 90.65 | 67.51  | 70.23  |
> | Ours with 5% feedback | 70.96   | 26.73    | 94.56      | 70.95 | 49.82 | 65.00   | 76.86     | 87.17   | 92.78 | 70.49  | 75.26  |
>
> 3. Moreover, we are also changing the previous comparison method to introduce human feedback and provide a stronger baseline. Specifically, we added an uncertain sample cache on the TDA to maintain the mean of uncertain samples. The experimental results are shown below. The experimental results indicate that the proposed method achieves superior performance. However, TDA may fail in certain cases due to its inefficient utilization of human feedback samples. Specifically, merely storing uncertain samples and their labels in the cache does not lead to performance improvement, possibly because the features of uncertain samples may contain data noise.
>
>
> | Method                | ImageNet | Average except ImageNet | FGVC  | Caltech101 | Cars  | DTD   | EuroSAT | Flower | Food101 | Pets  | SUN397 | UCF101 |
> |-----------------------|----------|---------|-------|------------|-------|-------|---------|--------|---------|-------|--------|--------|
> |TDA 5% Human Feedback     | 69.46    | 65.31   | 23.13 | 91.36      | 64.73 | 41.78 | 55.54   | 69.47  | 85.87   | 89.48 | 64.54  | 67.17  |
> | Ours with 5% feedback | 71.01| 70.96   | 26.73    | 94.56      | 70.95 | 49.82 | 65.00   | 76.86     | 87.17   | 92.78 | 70.49  | 75.26  |
> |TDA 15% Human Feedback    | 69.68    | 67.44   | 23.73 | 91.93      | 66.02 | 44.27 | 64.06   | 70.52  | 85.97   | 90.52 | 65.80  | 71.56  |
> | Ours with 15% feedback |71.83| 73.89|28.65 | 95.01 | 73.01 | 53.78 | 76.60 | 79.70 | 87.41 | 93.54 | 71.82 | 79.33 |
>
>
>
> [1] Active test time prompt learning in vision-language models
> ## Weakness 6. Amount of human feedback.
>
> 1. Firstly , based on your suggestions, we added experiments with human feedback ratios of 1% and 2%, and the results are shown in the following table. As can be seen from the table, the model can still achieve absolute performance improvement with only 1% feedback.
>
> | Human-Feedback | Acc   | Acc*  |
> |----------------|-------|-------|
> | 0%              | 70.68 | 70.68 |
> | 1%             | 70.77 | 71.52 |
> | 2%             | 70.79 | 72.26 |
>
>
> The number of feedback instances is influenced not only by the feedback ratio but also by the number of test samples. In practice, for most datasets, a 5% feedback ratio typically requires only about 200 feedback instances, averaging around 2 feedback instances per category. However, when the feedback ratio is too small, each category may receive as little as 0.1 feedback instances. For example, in the case of ImageNet, a 1% feedback ratio results in approximately 0.1 feedback instances per category. How to improve model performance when there are very few feedback times is another promising direction in the future. But it may be beyond the scope of this paper.

---

> ### Author Response · Authors · 2024-11-16
> **Official Response by Authors (3/3)**
>
> ## Weakness 7. Choice of hyperparameters
>
> During the experiment, to ensure a fair comparison with other methods, we adopted the same settings, specifically adjusting the model hyperparameters using the validation set. At the same time, we also reconducted the experiment, and the proposed method is basically quite robust to hyperparameters. This allows us to **achieve quite good performance on the test set without tuning parameters**. The experimental results are shown below.
>
> To better validate the sensitivity of our model to hyperparameters, we conducted systematic experiments and analyses with the following details:
>
> 1. On the Selection of Hyperparameter $\sigma$. We tested the values of $\sigma$ in the range \([0.0001, 0.001, 0.002, 0.004, 0.008, 0.02]\) while keeping other parameters fixed. The experimental results show that the model demonstrates strong robustness to different $\sigma$ values. The accuracy (acc) for each value of $\sigma$ is as follows:
>
>      | $\sigma$ | 0.0001 | 0.001 | 0.002 | 0.004 | 0.008 | 0.02  |
>      |------------|--------|-------|-------|-------|-------|-------|
>      | **Acc**    | 70.58  | 70.63 | 70.68 | 70.64 | 70.56 | 70.36 |
>
>    - These results indicate that the impact of $sigma$ on the model's performance is minimal, showcasing its robustness to this parameter.
>
> 2. **On the Selection of Hyperparameters $\eta$ and $\rho$:** Further experiments were conducted by fixing $\sigma$ and testing $\rho$ values in \([0.005, 0.01, 0.02, 0.03]\), while adjusting $\eta$ in \([0.2, 0.3, 0.4, 0.5]\). The results are summarized in the table below:
>
>      | $\eta \backslash \rho$ | 0.005  | 0.01   | 0.02   | 0.03   |
>      |--------------------------|--------|--------|--------|--------|
>      | **0.2**                  | 70.68  | 70.66  | 70.51  | 70.43  |
>      | **0.3**                  | 70.66  | 70.51  | 70.28  | 70.16  |
>      | **0.4**                  | 70.66  | 70.48  | 70.19  | 70.03  |
>      | **0.5**                  | 70.66  | 70.44  | 70.08  | 69.91  |
>
> These results further demonstrate the robustness of our model to the selection of $\eta$ and $\rho$, validating the stability of our proposed method. The above experiments were conducted using the ViT-B/16 backbone on the ImageNet validation set.
>
> Finally, since we can collect human feedback during testing, it may also be an interesting direction to perform real-time parameter adjustment with the help of human feedback labels in the future.
>
> ## Q1: Test Distribution Estimation
> The proposed method is a general framework, allowing the batch size during test-time adaptation to be arbitrary. In Eq.4 and Eq.6, the iterative updating equation are expressed in the form of tensor operations to maintain their generality. However, to ensure a fair comparison with baseline methods, we set the batch size to 1 in practice. Specifically, the proposed method estimates the data distribution of all samples in the test set in an online manner, i.e., $\mu$ and $\sigma$. There is no need to store the representations of test samples; instead, only their statistical features ($\mu$ and $\sigma$) are maintained. When a new sample is obtained, the stored statistical parameters ($\mu$ and $\sigma$) are dynamically iterated and updated based on the information from the current sample and previous information. $n$ is the number of the test sample, indicating which test sample it is.
>
> Instead of estimating a distribution for each sample, we estimate the data distribution of different categories for all test samples in an online manner, that is, given a sample, the estimated distribution is updated based on the information of the current sample.
>
> ## Q2. Selection of uncertain samples.
>
> ### Subquestion a. Why does confidence perform better than similarity in practice?
>
> Compared to similarity, confidence is more discriminative for uncertain samples. The reason is that **confidence implicitly takes into account the similarities of multiple different classes**, whereas the maximum similarity only considers the similarity to a single class. For example, when an image is close to both class 1 and class 2, the similarities to both classes might be high. However, after applying softmax, the confidence will be distributed between class 1 and class 2, making it easier for us to identify this uncertain sample.
>
> ### Subquestion b.  Confidence estimated from zero shot classifier?
>
> The confidence estimated is from zero shot classifier.  The key factor is that in practice, the zero-shot classifier is a relatively well-calibrated classifier. For example, on the ImageNet dataset, the error between its average confidence and classification accuracy is approximately (Expected Calibration Error, ECE) 1.51%, that is, the error between the confidence it gives and the true accuracy is less than 1.51%. Existing research has also shown that the pre-trained CLIP model is well-calibrated in the zero-shot setting[1].
>
> [1] Revisiting the Calibration of Modern Neural Networks

---

> ### Author Response · Authors · 2024-11-23
> **Clarification on Potential Misunderstandings and the Distributional Test-Time Adaptation Process**
>
> Because there may be some misunderstandings, we want to clarify any potential misunderstandings regarding our Distributional Test-Time Adaptation (DOTA) process. Below is a detailed explanation of the workflow:
>
> 1. **Global Distribution Estimation:**
>    Unlike estimating a separate distribution for each individual sample, our approach estimates the overall test data distribution across different classes during testing. This is achieved in an online manner, considering all test samples in a streaming manner.
>
> 2. **Class-wise Distribution Maintenance:**
>    For each class, we maintain distribution information in the form of a mean vector and a covariance matrix. As new samples are processed, these distribution parameters are updated iteratively by incorporating the information from previous estimated distribution and new samples.
>
> 3. **Efficient Iterative Updates:**
>    The equation used to update the class-wise distribution is fully vectorized, making it computationally efficient and independent of the number of processed samples. This ensures scalability and feasibility for various testing scenarios (single image TTA and multi image TTA).
>
> 4. **Bayesian Inference for Class Probabilities:**
>    With the updated mean vectors and covariance matrices for each class, we leverage Bayes' theorem to compute the class probabilities of each test sample. This enables robust and adaptive predictions during the testing phase.
>
> 5. **Versatility of the Proposed Approach:**
>    Our proposed method is versatile, functioning seamlessly in both single-image and multi-image scenarios. Its vectorized distribution estimation strategy, coupled with the maintenance of class-specific parameters, ensures robust and efficient adaptation to diverse test-time data distributions.
>
> We hope this explanation clarifies any doubts regarding our DOTA process and highlights its practical advantages. Thank you for your thoughtful feedback and the opportunity to further elucidate our approach.
>
> **If you have other confusions due to our unclear explanation, please do not hesitate to send us a message.**

---

> ### Author Response · Authors · 2024-11-25
> **By incorporating human feedback after model deployment, we can continuously improve the model's performance, even if we stop collecting labels at a later stage.**
>
> Thank you very much for your thoughtful response.
>
> We would be happy to provide an example to further explain why incorporating human feedback can be effective.
>
> Consider the situation where the model is already deployed. Without human feedback, the model's performance will remain static and will not improve.
>
> However, once human feedback is collected, the model’s performance can be further improved in terms of the standard ACC metric (standard acc means that when evaluating performance, without the help of human feedback labels). **In other words, after deployment, even if we stop collecting human labels at a certain stage, the model's performance of DOTA will still be better than it was before the feedback loop began.**  This could be particularly important for certain datasets, such as the EuroSAT dataset, where introducing 5% human feedback resulted in an increase in the model's standard accuracy from 57% to 65%.
>
> **At the same time, as the model’s performance improves, we can gradually reduce the amount of human feedback required by using a higher feedback ratio in the early stages of deployment and a lower ratio in the later stages.** When we stop collecting human feedback, the ratio of human feedback will continue to decrease as the number of test samples increases.
>
> We hope the above discussion addresses your concerns. If you have any further questions, we would be happy to continue the conversation.
>
> Finally, if you consider this an important task and scenario, it might be worthwhile to allow this paper to be accepted so that it can reach a wider audience and contribute to further improving performance of TTA with human feedback.

---

> ### Author Response · Authors · 2024-11-25
> **Further experiments to show the effectiveness of TTA with less human feedback on large-scale dataset.**
>
> Dear reviewer,
>
> Thank you for your constructive comments.
>
> We evaluate the impact of incorporating human feedback on model performance using a larger dataset (over 1 million test samples). Specifically, we introduce more human feedback during the early stages of model testing (the first 50,000 samples), but stop introducing feedback or updating the model in the later stages of testing. The experimental results are as follows. **It can be seen that when the model is adapted during testing with human feedback, the more test samples the model has in the future, the greater the benefits it brings, and the lower the cost of human feedback.**
>
> | Model                        | Performance (%) in terms of standard ACC |
> |------------------------------|------------------------------------------|
> | Original CLIP                | 70.14                                   |
> | DOTA without human feedback  | 70.89                                   |
> | Dota with Feedback rate at 0.75%       | 72.01                                   |
> | Dota with Feedback rate at 1%          | 72.44                                   |
> | Dota with Feedback rate at 2%          | 73.15                                   |
>
> Assuming the number of test samples increases by 10 times, and the model performance remains stable ( due to the data are independent and identically distributed), we can hypothesize that the corresponding performance could be as follows:
>
> | Model                        | Performance (%) in terms of standard ACC |
> |------------------------------|------------------------------------------|
> | Original CLIP                | 70.14                                   |
> | DOTA without human feedback  | 70.89                                   |
> | Dota with Feedback rate at 0.075%      | 72.01                                   |
> | Dota with Feedback rate at 0.1%        | 72.44                                   |
> |Dota with  Feedback rate at 0.2%        | 73.15                                   |
>
> These results suggest that incorporating human feedback, even in small feedback rate, is crucial for improving model performance and achieving higher levels of accuracy. We also acknowledge that there may be some trade-offs between the amount of human feedback introduced and the performance of the model, and that this may not be applicable in all scenarios.

---

> > ### Author Response · Authors · 2024-11-27
> >
> > Dear Reviewer, since November 27th is the final deadline for submitting the manuscript, I would like to know if you have any concerns about our manuscript that we could address through revisions to improve its score.

---

> ### Author Response · Authors · 2024-11-27
> **Further manuscript updates**
>
> Dear reviewer, we have update the manuscript with our latest discussion and promise to improve it further.

---

### Author Response · Authors · 2024-11-16
**Thank you for considering our revisions and valuable sugestiongs. We are looking forward to disscuss with you.**

Dear Reviewers, Area Chairs, Program Chairs and Senior Area Chairs,

We address the reviewers' concerns with the following updates and improvements:

1. **Hyperparameter Analysis**: Detailed exploration of hyperparameter settings.
2. **Additional Evaluation Metrics**. Clarified previous evaluation metrics and added new evaluation metrics
3. **Improved Baselines**: New baseline methods, covering diverse uncertain sample selection strategies and the latest methods (3 published in NeurIPS 2024 and 1 under review at ICLR 2025).
4. **Extended Analysis and Experiments**: More evaluation, including variations of TDA, the impact analysis of data stream order, and scenarios with reduced human feedback ratios, etc.
5. **Diverse CLIP Versions**: Integration of another version of CLIP models (contrastive learning models for pathology and text).
6. **More Datasets**: Use of additional datasets, particularly for pathology image classification.
7. **Point-by-Point Responses**: Comprehensive responses to specific reviewer comments.

Thank you for considering our revisions and valuable suggestion. **We appreciate your help in our work. If you have any other concerns, please feel free to contact us and we look forward to disscuss with you.** A new manuscript is also in preparation in which we will address the reviewers' concerns.

---

### Author Response · Authors · 2024-11-24
**Manuscript has been updated!**

Dear reviewers, I have updated the paper based on your feedback. Due to space constraints, some of the results have been included in the appendix.

---

### Author Response · Authors · 2024-11-28
**Response for Reviewer QFMN (Reviewer 1)**

## **For Reviewer QFMN (Reviewer 1):**

Dear reviewers,

We clarify two existing misunderstandings at the beginning of this web page to facilitate further discussion.


**On the benefit of incorporating human feedback after deployment:**

Incorporating human feedback after model deployment can effectively enhance the model's performance, even when we stop collecting labels at a later stage. To elaborate, consider the scenario where the model is already deployed. Without human feedback, its performance will remain static and unimproved. However, when human feedback is incorporated, the model's performance can improve on standard accuracy (evaluated without using human feedback labels).

**In other words, even if human label collection ceases at a certain stage post-deployment, the performance of DOTA will remain higher than it was prior to introducing the feedback loop.** For example, with the EuroSAT dataset, introducing 5% human feedback increased standard accuracy from 57% to 65%.

Furthermore, as the model’s performance improves, we can gradually reduce the volume of human feedback. A higher feedback ratio can be applied during the early deployment stages, with a lower ratio later. When human feedback collection stops, the ratio of feedback naturally decreases as the number of test samples increases. **This indicates that the benefit of incorporating human feedback becomes increasingly significant.**

To validate this, we conducted experiments on a large dataset (over 1 million test samples). In these experiments, we introduced more human feedback during the early stages (first 50,000 samples) and then ceased feedback collection and model updates during the later stages. Results demonstrate that incorporating human feedback leads to increasing benefits as the number of test samples grows while reducing the cost of feedback.

Experimental results:

| Model                        | Performance (%) in terms of standard ACC |
|------------------------------|------------------------------------------|
| Original CLIP                | 70.14                                   |
| DOTA without human feedback  | 70.89                                   |
| DOTA with Feedback rate at 0.75%       | 72.01                                   |
| DOTA with Feedback rate at 1%          | 72.44                                   |
| DOTA with Feedback rate at 2%          | 73.15                                   |

Hypothetically, assuming the number of test samples increases tenfold and performance remains stable (as the data usually are independent and identically distributed), the corresponding performance could be:

| Model                        | Performance (%) in terms of standard ACC |
|------------------------------|------------------------------------------|
| Original CLIP                | 70.14                                   |
| DOTA without human feedback  | 70.89                                   |
| DOTA with Feedback rate at 0.075%      | 72.01                                   |
| DOTA with Feedback rate at 0.1%        | 72.44                                   |
| DOTA with Feedback rate at 0.2%        | 73.15                                   |

These results highlight the critical role of human feedback, even at low feedback rates, in enhancing model performance and achieving greater accuracy. We acknowledge that there is a trade-off between the volume of human feedback and performance gains, and this approach may not apply universally.

---

> ### Comment · Reviewer_QFMN · 2024-12-01
> **Response to authors**
>
> Dear authors,
>
> Thank you for the clarifications and the additional experiments performed for lower feedback rate. From a bigger perspective, I still have conflicting thoughts on the effectiveness of having human feedback during TTA. I do believe it is a problem worth exploring. Yet, even with all the experiments you have presented, I don't see it being effective enough in terms of the effort (having a human labeler) to reward (accuracy gain) ratio, in general. Only in the case of Eurosat, it seems to be useful, the results are shown with 5\%. I still stand by the belief that 5\% and 15\% is too much labels to ask for. The paper focuses on this rate of feedback in most experiments. I do acknowledge the authors efforts in doing experiments for lower feedback rate. This only becomes part of the analysis done but not the main experiments nor is expected to in this process.
>
> After all this I am choosing to be positively inclined by increasing the score, primarily because it is a problem that is worth exploring and acknowledging the efforts in this direction.
>
> Some suggestions or perspectives, maybe beyond the purview of this work alone. From a practical perspective, what would be a good study on TTA with human feedback:
>
> 1. The focus of the work should be on trying to achieve significant gains with minimal feedback (1-2%). From a practical perspective, if  1% data is labeled, an overall gain in accuracy (ACC*) of about 3-4% would justify the labeling cost. (None of the results in this work show performance improvement in this order. I cannot comment on what is possible to achieve based on these observations. I am only intending to say what is practical to expect from an algorithm here).
> 2. A more practical metric is ACC*. Well, if one has asked the label, why should the prediction be used to calculate the performance. This also inherently measures the sample selection method also. Selecting harder or inaccurate samples for labeling would result in higher acc* compared to accuracy without human feedback.
> 3. How and when are the samples selected, especially in TTA where one does not have access to all data at once? This plays a major role and I believe it is crucial to design this process well.
>
> I appreciate the authors for putting forth this problem setting. But it is still far from a practical approach for TTA with human feedback.

---

> > ### Author Response · Authors · 2024-12-02
> > **Thank you for your valuable review.**
> >
> > Dear Reviewer,
> >
> > Thank you very much for your thoughtful and detailed feedback. We deeply appreciate the time and effort you have devoted to reviewing our work. Your insights are valuable, particularly your perspective on the effort-to-reward ratio (i.e., the trade-off between the effort required from human labelers and the accuracy improvements achieved).
> >
> > Our future research could focus on the following potential directions:
> >
> > 1.	**Achieving a higher effort-to-reward ratio**: By considering the entire testing process as a whole, we can explore strategies to improve the reward within a fixed effort budget. For instance, the accuracy gains from introducing human labels early in the testing process may outweigh those achieved later.
> >
> > 2.	**Enhancing evaluation metrics**: Introducing additional evaluation metrics that simultaneously account for both sample selection strategies and the model's learning capabilities.
> >
> > 3.	**Improving uncertain sample selection strategies**: Designing more effective strategies for detecting uncertain samples within test data streams, such as employing a caching mechanism to store uncertain samples for future selection.
> >
> > The issues you have highlighted are insightful, and we believe they represent promising directions for future research. Once again, we sincerely thank you for your valuable review.
> >
> > Best regards

---

### Author Response · Authors · 2024-11-28
**Response for Reviewer GKJd (Reviewer 4)**

Dear reviewers,

We clarify two existing misunderstandings at the beginning of this web page to facilitate further discussion.

## **For Reviewer GKJd (Reviewer 4):**

**Comparison with T3A:**

T3A is a classic test-time adaptation (TTA) method. However, if T3A retains all past samples, its repository grows with the number of test samples, leading to increased storage and computational costs. Specifically, the storage and computational complexity of T3A are \(O(n)\), where \(n\) is the number of test samples. In contrast, our method requires only the mean and covariance matrix, avoiding additional storage, and achieving \(O(1)\) complexity for both storage and computation. In addition, simply memorizing samples may not result in a good classifier, which is similar to the comparison between KNN and Gaussian discriminant analysis algorithms.

**On the negative impact of low-confidence samples:**

1. In fact, our initial version of the proposed method updated model parameters using only high-confidence samples, following the strategy used in T3A and TDA methods. However, the performance improvement was limited compared with TDA. The fundamental reason is that low-confidence predictions also contain information that is beneficial for model training. The experimental results are shown in the following table:

|                              | Caltech101 | Cars  | DTD   | EuroSAT | Flower102 | Food101 | Pets  | SUN397 | UCF101 | Mean      |
|------------------------------|------------|-------|-------|---------|-----------|---------|-------|--------|--------|-----------|
| TDA                          | 94.24      | 67.28 | 47.40 | 58.00   | 71.42     | 86.14   | 88.63 | 67.62  | 70.66  | 72.38     |
| DOTA with high-confidence only  | 94.10      | 68.08 | 45.70 | 58.60   | 72.06     | 88.06   | 89.47 | 69.90  | 68.80  | 72.75     |
| DOTA with all samples        | 94.32      | 69.48 | 47.87 | 57.65   | 74.67     | 87.02   | 91.69 | 69.70  | 69.01  | 73.49     |

2. **In some deployment scenarios, we cannot obtain a sufficient number of high-confidence samples to characterize the data distribution.** On the ImageNet test set, each class contains only 50 samples, insufficient for selecting a large number of high-confidence samples for adaptation. Utilizing information from low-confidence samples is crucial. Additionally, due to inherent CLIP biases, certain categories may naturally have low confidence, making high-confidence-only adaptation challenging.

3. Theoretically, our method’s objective function aligns with prior TTA methods, such as entropy minimization. These methods minimize cross-entropy loss between the predicted class probabilities.

4. This process resembles a single iteration of the Expectation-Maximization (EM) algorithm. Like EM, our method uses predicted probabilities of the old classifier to train the new classifier. Specifically, zero-shot classification corresponds to the expectation step, while estimating \(\{\mu_k, \Sigma_k\}\) represents the maximization step.

5. Intuitively, this estimation process can be seen as reweighting. Zero-shot predicted probabilities (typically well-calibrated [1]) serve as weights to adjust sample contributions across classes, mitigating inaccuracies in zero-shot predictions.

However, we also very acknowledge your perspective that, in some cases, using only high-confidence samples can also be a good choice, and it serves as a complementary approach to the method we proposed, i.e., the proposed method can also use this strategy, and as shown in the table above, the performance is still slightly improved compared to TDA.

[1] Revisiting the calibration of modern neural networks.

We deeply appreciate your thoughtful and constructive feedback. Thank you for guiding us in improving our work.

---

> ### Comment · Reviewer_GKJd · 2024-12-02
> **Thanks for Response**
>
> Dear authors,
>
> Thank you for the response and clarifications. This work does provide some new insights for TTA in vision-language models. Although there are some similarities with existing works like T3A/TDA, as they dynamically update the weights of the classifier/adapter and Dota updates the feature distribution for classification. I would like to increase my score to 6.

---

> > ### Author Response · Authors · 2024-12-02
> > **Thank you for your time and valuable review.**
> >
> > Dear Reviewer,
> >
> > Thank you for your thoughtful response and for taking the time to reconsider your score. We are glad to hear that you find our work valuable and that it provides new insights for Test-time Adaptation (TTA) in vision-language models. Once again, thank you for your time and valuable review.
> >
> > Best regards

---

### Meta-Review · Area_Chair_BF3y · 2024-12-18

**Metareview:**

This paper tackles the test-time adaptation problem for CLIP using an approach termed distributional test-time adaptation (DOTA). The approach has two main ideas: the first idea is to estimate and keep track of the distribution parameters (namely mean and variance) of test samples, and the second idea is to integrate the approach with a human-in-the-loop strategy, which asks human labelers to correct wrong zero-shot predictions when the model’s confidence is lower than a threshold. The reviewers generally appreciated the first idea of distributional adaptation but remained deeply concerned about the human-in-the-loop design, which does not bring significant improvement and lacks comparisons with related work. In the panel discussion, the reviewers specifically pointed out that the marginal improvement brought by the extra human labor cost is insufficient and thus do not give support for full acceptance. After reading the paper and the reviews, the AC agrees with the view that the human-in-the-loop design does not bring enough improvement and therefore needs further justifications. Since this is the first work that combines test-time adaptation with human-in-the-loop, it is important to set up a fair, convincing benchmark for future research. The AC recommends that the paper be rejected and encourages the authors to further refine the human-in-the-loop part.

**Additional Comments On Reviewer Discussion:**

In the rebuttal, the authors provided detailed information about the experiments, as requested by the reviewers. The reviewers also engaged actively in the discussions. However, the main concern about the human-in-the-loop benchmark remains unresolved, as discussed in the panel discussion.

---

### Decision · Program_Chairs · 2025-01-22

Reject